# DIVERSE GRAPH-BASED NEAREST NEIGHBOR SEARCH

## ABSTRACT

Nearest neighbor search is a fundamental data structure problem with many applications in machine learning, computer vision, recommendation systems and other fields. Although the main objective of the data structure is to quickly report data points that are closest to a given query, it has long been noted (Carbonell & Goldstein, 1998) that without additional constraints the reported answers can be redundant and/or duplicative. This issue is typically addressed in two stages: in the first stage, the algorithm retrieves a (large) number $r$ of points closest to the query, while in the second stage, the $r$ points are post-processed and a small subset is selected to maximize the desired diversity objective. Although popular, this method suffers from a fundamental efficiency bottleneck, as the set of points retrieved in the first stage often needs to be much larger than the final output.

In this paper we present provably efficient algorithms for approximate nearest neighbor search with diversity constraints that bypass this two stage process. Our algorithms are based on popular graph-based methods, which allows us to "piggyback" on the existing efficient implementations. These are the first graph-based algorithms for nearest neighbor search with diversity constraints. For data sets with low intrinsic dimension, our data structures report a diverse set of $k$ points approximately closest to the query, in time that only depends on $k$ and $\log \Delta$, where $\Delta$ is the ratio of the diameter to the closest pair distance in the data set. This bound is qualitatively similar to the best known bounds for standard (non-diverse) graph-based algorithms. Our experiments show that the search time of our algorithms is substantially lower than that using the standard two-stage approach.

## 1 INTRODUCTION

Nearest neighbor search is a classic data structure problem with many applications in machine learning, computer vision, recommendation systems and other areas (Shakhnarovich et al., 2006). It is defined as follows: given a set $P$ of $n$ points from some space $X$ equipped with a distance function $D(\cdot, \cdot)$, build a data structure that, given any query point $q \in X$, returns a point $p \in P$ that minimizes $D(q, p)$. In a more general version of the problem we are given a parameter $k$, and the goal is to report $k$ points in $P$ that are closest to $q$. In a typical scenario, the metric space $(X, D)$ is the $d$-dimensional space, and $D(p, q)$ is the Euclidean distance between points $p$ and $q$.

Since for high-dimensional point sets the known *exact* nearest neighbor search data structures are not efficient, several approximate versions of this problem have been formulated. A popular theoretical formulation relaxes the requirement that the query algorithm must return the exact closest point $p$, and instead allows it to output any point $p' \in P$ that is a $c$-approximate nearest neighbor of $q$ in $P$, i.e., $D(q, p') \leq cD(q, p)$. In empirical studies, the quality of the set of points reported by an approximate data structure is measured by its recall, i.e., the average fraction of the true $k$ nearest neighbors returned by the data structure.

Although maximizing the similarity of the reported points to the query is often the main objective, it has long been noted (Carbonell & Goldstein, 1998) that, without additional constraints, the reported answers are often redundant and/or duplicative. This is particularly important in applications such as recommendation systems or information retrieval, where many similar variants of the same product, product seller or document exist[1] To avoid reporting a long list of redundant answers, the search

---

[1]For example, an update to the search results listing algorithm implemented by Google in 2019 ensurers "no more than two pages from the same site" (Liaison, 2019).

problem is reformulated to ensure that the reported answers are sufficiently *diverse*, according to some metric $\rho$ (typically different from $D$). For example, the paper by Carbonell & Goldstein (1998) proposed to augment the search objective so that, in addition to minimizing the dissimilarity between the query and the reported set $S$, it also maximizes the pairwise dissimilarity $\rho$ between the elements in $S^2$. The paper stimulated the development of the rich area of *diversity-based reranking*, which became the dominant approach to this problem. The approach proceeds in two stages. In the first stage, the data structure retrieves $r$ points closest to the query, where $r$ can be much larger than the desired output $k$. In the second stage, the $r$ points are post-processed to maximize the diversity objective of the reported $k$ points.

Although popular, the reranking approach to diversifying nearest neighbor search suffers from a fundamental efficiency bottleneck, as the data structure needs to retrieve a large enough set to ensure that it contains the $k$ diverse points. In many scenarios, the number $r$ of points that need to be retrieved can be much larger than $k$ (see e.g., Figure 1 and the discussion in the experimental section). In the worst case, it might be necessary to set $r = \Omega(n)$ to ensure that the optimal set is found. This leads to the following algorithmic question:

> *Q: Is it possible to bypass the standard reranking pipeline by directly reporting the $k$ diverse points, in time that depends on $k$ and not $r$?*

This question has been studied in a sequence of papers (Abbar et al., 2013a;b). In particular (Abbar et al., 2013b) presented the following "diversified version" of the Locality-Sensitive Hashing (LSH) algorithm due to Indyk & Motwani (1998). Let $X$ contain all binary vectors of dimension $d$, the metric $D$ be the Hamming distance between points in $X$, and let $\rho$ be an arbitrary diversity metric on $X$. Furthermore, let $R > 0$ be the "search radius", $c > 1$ be the approximation factor, and $k$ be the target size of the output. Suppose that the data structure is given a query $q$ and let $S$ be the set of points in $P$ within distance $R$ from $q$. Define

$$\mathrm{div}(S) = \max_{S' \subset S, |S'| = \min(k, |S|)} \min_{p, p' \in S'} \rho(p, p')$$

to be the measure of the diversity of $S$. Then the output $S''$ of the data structure of (Abbar et al., 2013b) consists of $\min(k, |S|)$ points within distance $cR$ from $q$, such that

$$\mathrm{div}(S'') \geq \mathrm{div}(S)/6$$

The running time of the query procedure is $(d + k + \log n)^{O(1)} n^{1/c}$, while the space used by the data structure is at most $(d + k + \log n)^{O(1)} n^{1+1/c}$. Note that the dependence of the query time and space bounds on the data set size $n$ is the same as for the standard Hamming LSH algorithm of (Indyk & Motwani, 1998).

This result shows that the answer to the above question $Q$ is positive. However, the algorithm suffers from several limitations. First, the distance functions $D$ are limited to Hamming distance or its variants like the Jaccard similarity (Abbar et al., 2013a). Although it is plausible that the result could be extended to other distances that are supported by LSH functions, not all distance functions satisfy this constraint. Furthermore, the last decade has witnessed the development of a class of highly efficient algorithms that do not rely on LSH. These algorithms are "graph-based": the data structure consists of a graph between the points in $P$, and the query procedure performs greedy search over this graph to find points close to the query. Graph-based algorithms such as HNSW (Malkov & Yashunin, 2018), NGT (Iwasaki & Miyazaki, 2018), and DiskANN (Jayaram Subramanya et al., 2019) have become popular tools in practice, often topping Approximate Nearest Neighbor benchmarks (Aumueller et al., 2024). In addition, they are highly versatile, as they do not put any restrictions on the distance function $D$. This raises the following variant of above question:

> *Q': is it possible to adapt graph-based algorithms so that they directly report $k$ diverse points, in time that depends on $k$ and not $r$?*

**Our results.** In this paper we give a positive answer to this question, by designing a variant of the DiskANN algorithm that reports approximate nearest neighbors of a given query satisfying

---

[2]Technically, the paper used the notion of *similarity* as opposed to *dissimilarity*, so the objective was formulated in a dual manner. Please see the original paper for more details.

diversity constraints. Our theoretical analysis of these variants follow the setup of (Indyk & Xu, 2023). Specifically, we assume that the input point set $P$ has bounded *doubling dimension*[3] $d$, and that its aspect ratio (the ratio of the diameter to the closest pair distance) is at most $\Delta$. Under this assumption, we show that the query time of our data structures is polynomial in $k$, $\log n$ and $\log \Delta$.

To state our results formally, we need some notation. We say that a set $S$ of $k$ points from $X$ is *C-diverse* if $\min_{p,p' \in S'} \rho(p, p') \geq C$. We further generalize this notion to allow the set to contain at most $k' > 1$ points that are similar to each other. Specifically, we say that a set $S$ is $(k', C)$-*diverse* if for any $p \in S$ there are at most $k' - 1$ other points $p' \in S$ such that $\rho(p, p') < C$.

We consider two dual variants of the diverse nearest neighbor search problem, both of which use two approximation factors: $c > 1$ is the "dissimilarity" approximation factor with respect to $D$, and $a > 1$ is the "diversity" approximation factor with respect to $\rho$.

For a query $q$, let $S(q) = \{p_1, ..., p_k\}$ be a list of $k$ points from $P$, sorted according to their distance from $q$. We use $S(q)_i$ to denote the distance of $p_i$ from $q$. We drop the argument $q$ when its value is clear from the context.

- **Primal version:** Given a value $C$, find a set of $k$ points that are approximately closest to the query while maintaining the desired level of diversity $C$. Formally, for any $q \in X$, if OPT is a $(k', C)$-diverse set of $k$ points which minimizes $\mathsf{OPT_k}$, then the data structure outputs ALG that is $(k', C/a)$-diverse such that $\mathsf{ALG_k} \leq c \cdot \mathsf{OPT_k}$.

- **Dual version:** Given a radius $R$, find a set of $k$ points that approximately lie within the radius $R$, while maximizing the diversity. Formally, for any $q \in X$, let $B_P(q, R)$ be the set of points in $P$ within distance $R$ from $q$ and let OPT be a $(k', C)$-diverse set of $k^* = \min(k, |B_P(q, R)|)$ points from $B_P(q, R)$ that maximizes $C$. Then the data structure outputs ALG of size $k^*$ that is $(k', C/a)$-diverse such that $\mathsf{ALG_{k^*}} \leq cR$.

Note that the dual version is analogous to the problem addressed in the prior work (Abbar et al., 2013b) described in the introduction.

Our main theoretical result is captured by the following theorem, which specifies the approximation and running time guarantees for our algorithm solving the primal version of the diverse nearest neighbor problem.

**Theorem 1.1.** *Let* $\mathsf{OPT} = \{p_1^*, ..., p_k^*\}$ *be a* $(k', C)$-*diverse solution that minimizes* $\mathsf{OPT_k}$. *Given the graph constructed by Algorithm 1, the search Algorithm 2 finds a* $(k', C/12)$-*diverse solution* ALG *with* $\mathsf{ALG_k} \leq \left(\frac{\alpha+1}{\alpha-1} + \epsilon\right) \mathsf{OPT_k}$ *in* $O\left(k \log_\alpha \frac{\Delta}{\epsilon}\right)$ *steps, where each step takes* $O\left((k^3/k')(8\alpha)^d \log \Delta\right)$ *time. The data structure uses space* $O(n(k/k')(8\alpha)^d \log \Delta)$.

We note that the approximation factor with respect to $D$, as well as the running time bounds, are essentially the same as the bounds obtained in (Indyk & Xu, 2023) for a "slow-preprocessing" variant of the DiskANN algorithm. The main difference is that the bound in (Indyk & Xu, 2023) does not depend on $k$ or $k'$, as these parameters do not exist in the standard nearest neighbor formulation.

From the practical perspective, an important special case is the "colorful" version of the problem, where the diversity metric $\rho$ is *uniform*. That is, each point $p$ has a "color" (e.g., the seller id, the website id, etc.) denoted by $col[p]$, while the metric $\rho$ is such that $\rho(p_i, p_j) = 0$ for $col[p_i] = col[p_j]$ and $\rho(p_i, p_j) = 1$ otherwise. This is the version that is solved by the practical implementation of Algorithms 1 and 2. Note that the approximation factor w.r.t. $\rho$ plays no role in this setting, as all distances are either zero or non-zero.

We also give an improved analysis of Algorithm 2 for the case where $k' = 1$ (Theorem B.1). Specifically, we show an improved bound on the number of steps (by a factor of $k$); also we obtain an "entrywise" guarantee, where $\mathsf{ALG_i} \leq O(\mathsf{OPT_i})$ for every $i = 1 \ldots k$, not just $i = k$. Finally, we analyze Algorithm 3 for the dual version of the problem (assuming $k' = 1$) and give essentially the same complexity and approximation bounds as for the primal version. Compared to the prior, LSH-based algorithm for the dual version given in Abbar et al. (2013b), our algorithm has exponential dependence on the doubling dimension (similarly to other algorithms for this setting (Indyk & Xu,

---

[3]Doubling dimension is a measure of the intrinsic dimensionality of the pointset - see Preliminaries for the formal definition.

2023)), but avoids the polynomial dependence on $n$ (which is standard for LSH-based algorithms). In addition, our algorithm works for arbitrary metric spaces, not only the "LSH-able" ones.

**Experimental results.** For our experiments, we adapt our algorithms in two ways. First, we devise fast heuristic approximations of the graph construction algorithm (this is much like the differences between the fast- and slow-preprocessing algorithms in DiskANN (Jayaram Subramanya et al., 2019; Indyk & Xu, 2023)). Second, we restrict our implementation to cater to the colorful case. As one can see from the plots in Figures 2 and 3, both the new indexing and the search methods play a crucial role in improving the overall search quality. For example, to achieve 95% recall@100 for the product dataset, the baseline reranking approach retrieving $r \gg k$ nearest neighbors followed by post-processing has latency upwards of 8ms, while the improved search algorithm alone brings it down to approximately 5ms. Making both indexing and search diverse further brings this down to around 1.5ms, resulting in an improvement upwards of **5X**.

## 2  PRELIMINARIES

Let $(X, D)$ be the underlying metric space, with distance function $D$. We use $B_D(p, r)$ to denote a ball centered at $p$ with radius $r$, i.e., $B_D(p, r) = \{u \in X : D(u, p) < r\}$. Similarly, the ball $B_\rho(p, r)$ is defined. We will drop the subscript $D$ if the metric is clear from the context.

We have a point set $P$ with $n$ points $p_1, ... p_n$. We say a point set $P$ has *doubling dimension* $d$ if for any point $p$ and radius $r$, the set $B(p, 2r) \cap P$ can be covered by at most $2^d$ balls with radius $r$.

**Lemma 2.1.** *Consider any point set $P \subset X$ with doubling dimension $d$. For any ball $B(p, r)$ centered at some point $p \in P$ with radius $r$ and a constant $k$, we can cover $B(p, r) \cap P$ using at most $m \leq O(k^d)$ balls with radius smaller than $r/k$, i.e. $B(p, r) \cap P \subset \bigcup_{i=1}^{m} B(p_i, r/k)$ for some $p_1 \ldots p_m \in X$.*

We define $\Delta = \frac{D_{max}}{D_{min}}$ to be the *aspect ratio* of the point set $P$ where $D_{max}(D_{min}$, resp.) represents the maximal (minimal, resp.) distance between any two points in the point set $P$.

The following definition recaps the discussion in the introduction.

**Definition 2.2** (($k', C$)-diverse)**.** *Let $S$ be a point set in a metric space $(X, \rho)$ where $\rho(p_1, p_2)$ measures the diversity of two points $p_1, p_2$. We say $S$ is $(k', C)$-diverse if for any point $p \in S$, we have $|B_\rho(p, C) \cap S| \leq k'$.*

Let $\mathsf{ALG} = \{p_1, ..., p_k\}$ and $\mathsf{OPT} = \{p_1^*, ..., p_k^*\}$ be any two sets consisting of $k$ points. We write $\mathsf{ALG} \leq \mathsf{OPT}$ if for any $i$, $\mathsf{ALG_i} \leq \mathsf{OPT_i}$, where (as defined earlier) $\mathsf{ALG}_i$ (respectively $\mathsf{OPT}_i$) is the distance of the $i$th closest point in $\mathsf{ALG}$ (respectively $\mathsf{OPT}$) to the query $q$.

In the experimental section, we will consider a simplified version of the problem where the diversity metric $\rho$ is uniform. That is, we use $col[p]$ to denote the *color* of a point $p$, and define $\rho(p_i, p_j) = 0$ for $col[p_i] = col[p_j]$ and $\rho(p_i, p_j) = 1$ otherwise.

**Definition 2.3** (($k'$-colorful)**.** *Let $P$ be a point set. For each $p \in P$, we use $col[p]$ to denote its color. A set of points $\mathsf{ALG} = \{p_1, ..., p_k\}$ is $k'$-colorful if within the multi-set $\{col[p_1], ..., col[p_k]\}$, no color appears more than $k'$ times.*

For simplicity, we assume $k$ is a multiple of $k'$.

## 3  ALGORITHMS

In this section we describe our algorithms.

**The preprocessing algorithm.** The indexing algorithm, which is the same for both the primal and dual versions of the problem, is shown in Algorithm 1. Line 12 of the algorithm uses the greedy algorithm of Gonzalez (1985), defined below.

**Gonzales' greedy algorithm.** Given a set of $n$ points and a parameter $m$, the algorithm picks $m$ points as follows. The first point is chosen arbitrarily. Then, in each of $m - 1$ steps, the algorithm picks the point whose minimum distance w.r.t. $\rho$ to the currently chosen points is maximized. It is

---

**Algorithm 1** Indexing algorithm for diverse NN

---

1: **Input**: A set of $n$ points $P = \{p_1, ..., p_n\}$. $k$ is the size of the output. $k'$ is the parameter in the diversity definition. $\alpha$ is the parameter used for pruning.
2: **Output**: A directed graph $G = (V, E)$ where $V = \{1, ..., n\}$ are associated with the point set $P$.
3: **for** each point $p \in P$ **do**
4:     Sort all points $u \in P$ based on their distance from $p$ and put them in a list $\mathcal{L}$ in that order
5:     **while** $\mathcal{L}$ is not empty **do**
6:         $u \leftarrow \underset{u \in \mathcal{L}}{\operatorname{argmin}} D(u, p)$
7:         Initialize $\mathsf{bag}[u] \leftarrow \{u\}$
8:         **for** each point $v \in \mathcal{L}$ in order **do**
9:             **if** $D(u, v) \leq D(p, u)/(2\alpha)$ **then**
10:                 $\mathsf{bag}[u] \leftarrow \mathsf{bag}[u] \cup v$
11:                 remove $v$ from $\mathcal{L}$
12:         $\mathsf{rep}[u] \leftarrow$ use the greedy algorithm of Gonzales to choose $k/k'$ points in $\mathsf{bag}[u]$ to approximately maximize the minimum pairwise diversity distance.
13:         add edges from $p$ to $\mathsf{rep}[u]$
14:         Remove $u$ from $\mathcal{L}$

---

known Gonzalez (1985) that this algorithm provides a 2-approximation for the problem of picking a subset of size $m$ which maximizes the minimum pairwise diversity distance between the picked points. Moreover, the picked set has an anti-cover property which we will discuss in Proposition 3.4.

**Primal Search Algorithm.** Algorithm 2 shows the search algorithm for the primal version of diverse nearest neighbor. The algorithm has a different condition for $k' = 1$ as in this case we can slightly improve the performance of the algorithm as shown in Section B.1. The general case of the algorithm is analyzed in Section 3.1. The initialization step of line 3, can be done using the following algorithm.

---

**Algorithm 2** Search algorithm for primal diverse NN

---

1: **Input**: A graph $G = (V, E)$ with $N_{out}(p)$ be the out edges of $p$, query $q$, optimization step $T$, diversity lower bound $C$.
2: **Output**: A set of $k$ points $\mathsf{ALG}$.
3: Initialize $\mathsf{ALG} = \{p_1, ..., p_k\}$ to be the $k$ points satisfying $(k', 0.1C)$-diverse constraint using the initialization step proved in Lemma 3.2.
4: **for** $i = 1$ to $T$ **do**
5:     $U \leftarrow \underset{p \in \mathsf{ALG}}{\bigcup} (N_{out}(p) \cup p)$ and sort $U$ based on their distance from $q$
6:     **if** $k' = 1$ **then**
7:         $\mathsf{ALG} = \varnothing$
8:     **else**
9:         $\mathsf{ALG} \leftarrow$ the closest $k - 1$ points in $\mathsf{ALG}$
10:     **for** each point $u \in U$ in order **do**
11:         **if** $\mathsf{ALG} \bigcup u$ is $(k', 0.1C)$-diverse **then**
12:             $\mathsf{ALG} \leftarrow \mathsf{ALG} \cup u$
13:         **if** $|\mathsf{ALG}| = k$ **then**
14:             Break
15: **Return** $\mathsf{ALG}$

---

**The initialization step.** Given a set $P$ of $n$ points equipped with metric distance $\rho$, and parameters $k'$ and $k$, and lower bound diversity $C$, the goal is to pick a subset $S \subseteq P$ of size $k$ which is $(k', C/4)$ diverse or otherwise output that no $(k', C)$-diverse subset $S$ exists. We use the following algorithm

- Initialize $\mathsf{SOL} = \emptyset$

- While there exists a point $p \in P$ such that the ball $B = B_\rho(p, C/4)$ has $k'$ points in it, (i.e., $|B \cap P| > k'$),

    - Add an arbitrary subset of $B \cap P$ of size $k'$ to $\mathsf{SOL}$.
    - Remove all points in $2B = B_\rho(p, C/2)$ from $P$.

- Add all remaining points in $P$ to SOL.
- If $|\text{SOL}| \geq k$, return an arbitrary subset of it of size $k$, otherwise return 'no solution'.

**Lemma 3.1.** *If $P$ has a subset* OPT *of size $k$ that is $(k', C)$-diverse, our initialization algorithm finds a $(k', C/4)$-diverse subset of size $k$. (Proof in Appendix A)*

**Dual Search Algorithm.** Algorithm 3 shows the search algorithm for the dual version of the diverse nearest neighbor problem. We provide the analysis in Section B.2.

### 3.1 ANALYSIS OF THE PRIMAL DIVERSE NN ALGORITHM

In this section we prove Theorem 1.1 that gives the approximation and running time guarantees for Algorithm 1 and Algorithm 2.

**Lemma 3.2.** *The graph constructed by Algorithm 1 has degree limit $O((k/k')(8\alpha)^d \log \Delta)$.*

*Proof.* Let's first bound the number of points not removed by others, then according to Line 12 in Algorithm 1, the degree bound will be that times $k/k'$.

We use $\text{Ring}(p, r_1, r_2)$ to denote the points whose distance from $p$ is larger than $r_1$ but smaller than $r_2$. For each $i \in [\log_2 \Delta]$, we consider the $\text{Ring}(p, D_{max}/2^i, D_{max}/2^{i-1})$ separately. According to Lemma 2.1, we can cover $\text{Ring}(p, D_{max}/2^i, D_{max}/2^{i-1}) \cap P$ using at most $m \leq O((8\alpha)^d)$ small balls with radius $\frac{D_{max}}{2^{i+2}\alpha}$. According to the pruning criteria in Line 9, within each small ball, there will be at most one point remaining. This establishes the degree bound of $O((k/k')(8\alpha)^d \log \Delta)$. $\qquad\square$

**Lemma 3.3.** *Suppose* OPT $= \{p_1^*, ..., p_k^*\}$ *is a $(k', C)$-diverse solution with minimized* $\text{OPT}_k$ *and* ALG $= \{p_1, ..., p_k\}$ *be the current solution (ordered by distance from $q$). If $p_k \notin$ OPT*, there *exists a point $p^* \in$ OPT $\setminus$ ALG *such that $|B_\rho(p^*, C/2) \cap (\text{ALG} \setminus p_k)| < k'$ and* ALG $\setminus p_k \bigcup p^*$ *is $(k', C/4)$-diverse.*

*Proof.* We use $B_\rho(p, r)$ to denote the ball in the $(X, \rho)$ metric space. Because $p_k \notin$ OPT, we have $\overline{\text{OPT}} = \text{OPT} \setminus \text{ALG} \neq \varnothing$. We repeatedly perform the following operation until $\overline{\text{OPT}}$ gets empty: select a point $p$ from $\overline{\text{OPT}}$, get $z = B_\rho(p, C/2) \cap \overline{\text{OPT}}$, and remove $z$ from $\overline{\text{OPT}}$. By doing this, we can get a list of points $\{p_1^*, ..., p_m^*\}$ and a partition of OPT $\setminus$ ALG $= z_1 \cup z_2... \cup z_m$. By definition, we have the following properties:

- $\{p_1^*, ..., p_m^*\} \cap \text{ALG} = \varnothing$

- $z_i \cap z_j = \varnothing$ for $i \neq j$

- $\sum_i |z_i| = |\text{OPT} \setminus \text{ALG}| = |\text{ALG} \setminus \text{OPT}|$

Now let $w_i = B_\rho(p_i^*, C/2) \cap (\text{ALG} \setminus p_k \setminus \text{OPT})$. Because all the $B_\rho(p_i^*, C/2)$ balls are disjoint, $\sum_i |w_i| \leq |\text{ALG} \setminus p_k \setminus \text{OPT}| < |\text{OPT} \setminus \text{ALG}| = \sum_i |z_i|$, there must exist an $i$ such that $|w_i| < |z_i|$. For that $i$, we have that $|B_\rho(p_i^*, C/2) \cap (\text{ALG} \setminus p_k)|$ is equal to

$$=|B_\rho(p_i^*, C/2) \cap (\text{ALG} \cap \text{OPT})| + |B_\rho(p_i^*, C/2) \cap (\text{ALG} \setminus p_k \setminus \text{OPT})| \qquad \text{(Because } p_k \notin \text{OPT)}$$
$$=|B_\rho(p_i^*, C/2) \cap (\text{ALG} \cap \text{OPT})| + |w_i| < |B_\rho(p_i^*, C/2) \cap (\text{ALG} \cap \text{OPT})| + |z_i|$$
$$\leq|B_\rho(p_i^*, C/2) \cap (\text{ALG} \cap \text{OPT})| + |B_\rho(p_i^*, C/2) \cap (\text{OPT} \setminus \text{ALG})| = |B_\rho(p_i^*, C/2) \cap \text{OPT}| \leq k'$$

Therefore, we get $B_\rho(p_i^*, C/2) \cap (\text{ALG} \setminus p_k) < k'$. Now, for any point $p \in B_\rho(p_i^*, C/4)$, $|B_\rho(p, C/4) \cap (\text{ALG} \setminus p_k)| \leq |B_\rho(p_i^*, C/2) \cap (\text{ALG} \setminus p_k)| < k'$, so we know that ALG $\setminus p_k \cup p_i^*$ is $(k', C/4)$-diverse.

$\qquad\square$

The following is the well-known anti-cover property of the greedy algorithm of Gonzales whose proof we include in Section A for the sake of completeness.

**Proposition 3.4.** *In Line 12 of Algorithm 1, let* $\mathsf{rep}[u]$ *be the output of greedily choosing* $k/k'$ *points in* $\mathsf{bag}[u]$ *maximizing pairwise diversity. If a point* $p \in \mathsf{bag}[u] \setminus \mathsf{rep}[u]$, *we have* $\min\limits_{v \in \mathsf{rep}[u]} \rho(p, v) \leq \min\limits_{v_1, v_2 \in \mathsf{rep}[u]} \rho(v_1, v_2)$. *(Proof in Appendix A)*

**Lemma 3.5.** *There always exists a point* $p'$ *connected from some point* $w \in \mathsf{ALG}$ *such that*

1. $\mathsf{ALG} \setminus p_k \bigcup p'$ *is* $(k', C/12)$*-diverse*

2. $D(p', q) \leq D(p_k, q)/\alpha + \mathsf{OPT_k}(1 + 1/\alpha)$

*Proof.* According to Lemma 3.3, for any current solution $\mathsf{ALG}$ with $p_k \notin \mathsf{OPT}$, there exists a point $p^* \in \mathsf{OPT} \setminus \mathsf{ALG}$ such that $\mathsf{ALG} \setminus p_k \cup p^*$ is $(k', C/4)$-diverse. Let $w \in \mathsf{ALG}$ be the closest point to $p^*$. If there exists an edge from $w$ to $p^*$, replacing $p_k$ with $p^*$ is a potential update. We set $p' = p^*$ and $D(p', q) \leq \mathsf{OPT_k}$ satisfies the distance upper bound above.

Otherwise, we let $u$ be the point where $p^* \in \mathsf{bag}[u]$ but not selected into $\mathsf{rep}[u]$. For any point $p' \in \mathsf{bag}[u]$, $D(p', u) < D(w, u)/(2\alpha)$, so $D(p', p^*) < D(w, u)/\alpha < D(w, p^*)$. This means that all points in $\mathsf{bag}[u]$ are closer to $p^*$ than $w$, so they can't belong to $\mathsf{ALG}$. In the following, we consider two cases depending on whether $\min\limits_{v \in \mathsf{rep}[u]} \rho(p^*, v) \geq C/3$. In each case, we will find a desired $p' \in \mathsf{rep}[u]$ and it is connected to $w$.

1. $\min\limits_{v \in \mathsf{rep}[u]} \rho(p^*, v) < C/3$: In this case, there exists another point $p' \in \mathsf{rep}[u]$ with $D(p^*, p') \leq D(p^*, u) + D(u, p') \leq D(w, u)/\alpha$ and $\rho(p^*, p') < C/3$. Because $|B_\rho(p^*, C/2) \bigcap (\mathsf{ALG} \setminus p_k)| < k'$, we have $|B_\rho(p', C/6) \bigcap (\mathsf{ALG} \setminus p_k)| \subseteq |B_\rho(p^*, C/2) \bigcap (\mathsf{ALG} \setminus p_k)| < k'$, so the addition of such $p'$ satisfies that $\mathsf{ALG} \setminus p_k \cup p'$ is $(k', C/12)$-diverse.

2. $\min\limits_{v \in \mathsf{rep}[u]} \rho(p^*, v) \geq C/3$: In this case, according to Proposition 3.4, we have $\mathsf{rep}[u] = \{z_1, ..., z_{k/k'}\} \subseteq B(u, D(u, w)/(2\alpha))$ all with diversity distance at least $C/3$ from each other. Therefore, for any $p_i \in \mathsf{ALG} \setminus p_k$, there can't exist two $z_j$ and $z_{j'}$ s.t. $\rho(p_i, z_j) < C/6$ and $\rho(p_i, z_{j'}) < C/6$. By a counting argument, we can find at least one $z_i$ s.t. $|B_\rho(z_i, C/6) \cap (\mathsf{ALG} \setminus p_k)| < k'$. Finally, we let $p' = z_i$ where $\mathsf{ALG} \setminus p_k \cup p'$ is $(k', C/12)$-diverse.

We have proved that the $p'$ we found satisfies the $(k', C/12)$-diverse criteria. Now we will bound its distance upper bound.

$$
\begin{aligned}
D(p', q) &\leq D(p^*, q) + D(p', p^*) \leq D(p^*, q) + D(p', u) + D(p^*, u) \\
&\leq D(p^*, q) + D(w, u)/(2\alpha) + D(w, u)/(2\alpha) \quad &\text{(Line 9 in Algorithm 1)} \\
&\leq D(p^*, q) + D(w, u)/\alpha \\
&\leq D(p^*, q) + D(w, p^*)/\alpha \quad &\text{(Because } u \text{ is ordered earlier than } p^*) \\
&\leq D(p^*, q) + D(w, q)/\alpha + D(p^*, q)/\alpha \leq D(p_k, q)/\alpha + \mathsf{OPT_k}(1 + 1/\alpha)
\end{aligned}
$$

$\square$

*Proof of Theorem 1.1.* Regarding the running time, the total number of edges connected from any point in $\mathsf{ALG}$ is bounded by $|U| \leq O((k^2/k')(8\alpha)^d \log \Delta)$. In each step, the algorithm first sorts all these edges and then checks whether each of them can be added to the new $\mathsf{ALG}$ set. The total time spent per step is $O(k|U| + |U| \log |U|)$. Usually, we assume $k \gg \log |U|$, and we can have the overall time complexity to be $O\left((k^3/k')(8\alpha)^d \log \Delta\right)$ per step.

To analyze the approximation ratio, at time step $t$, we use $\mathsf{ALG^t} = \{p_1^t, ..., p_k^t\}$ to denote the current unordered solution. We denote $\mathsf{ALG_k^t} = \max\limits_{i \in [k]} D(p_i^t, q)$. According to Algorithm 2 and Lemma 3.5, if $p_i$ is updated at time step $t$, we have $D(p_i^t, q) \leq D(p_i^{t-1}, q)/\alpha + \mathsf{OPT_k}(1 + 1/\alpha)$. By an induction argument, if a point $p_i$ is updated by $t$ times at the end of time step $T$, we have $D(p_i^T, q) \leq \frac{D(p_i^0, q)}{\alpha^t} + \frac{\alpha+1}{\alpha-1}\mathsf{OPT_k}$.

We now prove that $\mathsf{ALG}_k^T \leq \max_i \frac{D(p_i^0, q)}{\alpha^{T/k}} + \frac{\alpha+1}{\alpha-1}\mathsf{OPT}_k$. Let $i \in [k]$ be the index achieving the maximal distance upper bound. For the sake of contradiction, if $\mathsf{ALG}_k^T > \frac{D(p_i^0, q)}{\alpha^{T/k}} + \frac{\alpha+1}{\alpha-1}\mathsf{OPT}_k$, this means that $p_i^T$ was updated for at most $T/k - 1$ times. By a counting argument, there exists another index $j$ which was updated for at least $T/k + 1$ times. However, at the time $t$ when $p_j^t$ was already updated for $T/k$ times, $D(p_j^t, q) \leq \frac{D(p_j^0, q)}{\alpha^{T/k}} + \frac{\alpha+1}{\alpha-1}\mathsf{OPT}_k < \mathsf{ALG}_k^T \leq \mathsf{ALG}_k^t$, so the algorithm wouldn't have chosen $p_j^t$ to optimize cause it couldn't have the maximal distance at that time, leading to a contradiction. Therefore, we prove that $\mathsf{ALG}_k^T \leq \max_i \frac{D(p_i^0, q)}{\alpha^{T/k}} + \frac{\alpha+1}{\alpha-1}\mathsf{OPT}_k$.

Now we consider the following three cases depending on the value of the maximal $D(p_i^0, q)$. The case analysis here is similar to the proof in Theorem 3.4 from (Indyk & Xu, 2023).

Case 1: $D(p_i^0, q) > 2D_{max}$. Let $p_k^*$ be the point having the maximal distance from $q$ in an optimal solution OPT. We know that for any $p_i^0$, we have $D(p_k^*, q) \geq D(p_i^0, q) - D(p_i^0, p_k^*) \geq D(p_i^0, q) - D_{max} \geq D(p_i^0, q)/2$. Therefore, the approximation ratio after $T$ optimization steps is upper bounded by $\frac{\mathsf{ALG}_k^T}{D(p_k^*, q)} \leq \frac{D(p_i^0, q)}{D(p_k^*, q)\alpha^{T/k}} + \frac{\alpha+1}{\alpha-1} \leq \frac{2}{\alpha^{T/k}} + \frac{\alpha+1}{\alpha-1}$. A simple calculation shows that we can get a $\left(\frac{\alpha+1}{\alpha-1} + \epsilon\right)$ approximate solution in $O(k\log_\alpha \frac{2}{\epsilon})$ steps.

Case 2: $D(p_i^0, q) \leq 2D_{max}$ and $\mathsf{OPT}_k > \frac{\alpha-1}{4(\alpha+1)}D_{min}$. To satisfy $\frac{D(p_i^0, q)}{\alpha^{T/k}} + \frac{\alpha+1}{\alpha-1}\mathsf{OPT}_k \leq \left(\frac{\alpha+1}{\alpha-1} + \epsilon\right)\mathsf{OPT}_k$, we need $\frac{D(p_i^0, q)}{\alpha^{T/k}} \leq \epsilon\mathsf{OPT}_k$. Applying the lower bound $\mathsf{OPT}_k \geq \frac{\alpha-1}{4(\alpha+1)}D_{min}$, we can get that $T \geq k\log_\alpha \frac{2(\alpha+1)\Delta}{(\alpha-1)\epsilon}$ suffices.

Case 3: $D(p_i^0, q) \leq 2D_{max}$ and $\mathsf{OPT}_k \leq \frac{\alpha-1}{4(\alpha+1)}D_{min}$. In this case, we must have $k = 1$, because otherwise $D(p_k^*, p_1^*) \leq 2D(p_k^*, q) < D_{min}$,violating the definition of $D_{min}$. Suppose $k = 1$ and the problem degenerates to the standard nearest neighbor search problem. After $T$ optimization steps, if $p_1^T$ is still not the exact nearest neighbor, we have $D(p_1^T, q) \geq D(p_1^T, p_1^*) - \mathsf{OPT}_1 \geq \frac{D_{min}}{2}$. Applying the upper bound of $D(p_1^T, q)$ and $\mathsf{OPT}_1$, we have $\frac{D_{min}}{2} \leq D(p_1^T, q) \leq \frac{D(p_1^0, q)}{\alpha^T} + \frac{\alpha+1}{\alpha-1}\mathsf{OPT}_1 \leq \frac{D(p_1^0, q)}{\alpha^T} + \frac{D_{min}}{4}$. This can happen only if $T \leq \log_\alpha \frac{\Delta}{8}$. $\qquad \square$

## 4 EXPERIMENTS

In this section we provide an empirical evaluation of our algorithm. We focus on the special case of the $k'$-colorful nearest neighbor problem as in Definition 2.3. Recall that in this setting, we use $col[p]$ to denote the color of a point $p$, and we define $\rho(p_i, p_j) = 0$ for $col[p_i] = col[p_j]$ and $\rho(p_i, p_j) = 1$ otherwise. In other words, we seek $k$ nearest neighbors, such that no more than $k'$ belong to any single color. Although restrictive, this case is of great practical interest in many settings, including shopping and search. In both of these applications, the data points represent products (resp. documents) and a color of a vector corresponds to seller (resp. domain) of the product. It is then desirable to output results from a diverse set of sellers or domains (Liaison, 2019). Intuitively, displaying diverse results would lead to increased competition between the sellers, and also simultaneously higher click probabilities, thereby leading to increase in revenue of the exchange.

For our experiments, we adapt our algorithms from Section 3 in two ways: one, we devise fast heuristic approximations of the graph construction algorithm (this is much like the differences between the fast- and slow-preprocessing algorithms in DiskANN (Jayaram Subramanya et al., 2019; Indyk & Xu, 2023)), and second, we restrict our implementation to cater to the special case of the $k'$-colorful version of the problem as defined in Definition 2.3. The pseudo-code of our efficient algorithms are described in Appendix C. All experiments were run on a Linux Machine with AMD Ryzen Threadripper 3960X 24-Core Processor CPU's @ 2.3GHz with 48 vCPUs and 250 GB RAM. All query throughput and latency measurements are reported for runs with 48 threads.

### 4.1 DATASETS AND ALGORITHMS

We consider three datasets for evaluation: one real-world dataset and two semi-synthetic datasets.

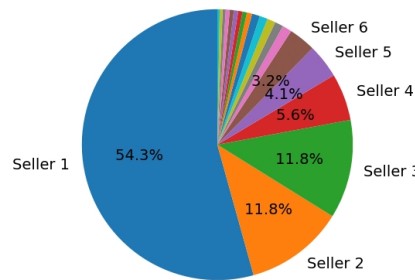

*Real-world dataset:* Our real world data set comprises of 64-dimensional vector embeddings of different products from a large advertisement corpus. Each product/vector is additionally associated with a *seller*, which becomes its color in our setting. There are 20 million base vectors and 5000 query vectors. The fraction of products corresponding to the top 20 sellers is shown in Figure 1. As shown in the figure, a small number of sellers constitutes more than 90% of the data, motivating the need for enforcing diversity in the search results.

Figure 1: Seller distribution in real-world data set.

*Semi-synthetic dataset:* We also consider the publicly available real-world Arxiv dataset (Embeddings, 2024) which contains OpenAI embeddings of around 2 million paper abstracts into 1536 dimensional vectors and the classical SIFT dataset of 1M vectors in 128 dimensions. These datasets do not contain any color information, so we synthetically add this information into the data set. Specifically, we generate the color information as follows: for each vector, with probability 0.9, we assign a color selected from the set $\{1, 2, 3\}$ uniformly at random, and with 0.1 probability we assign a color selected uniformly at random from the set $\{4, \ldots, 1000\}$. Therefore the number of distinct colors is at most 1000 in this data set. For the SIFT dataset, we sampled one dominant colors with probability 0.8 and had a uniform distribution over 999 other colors with probability 0.2.

As for algorithms, since our algorithms are enhancements of the DiskANN algorithm, we use that as a natural baseline to compare against.

**Standard DiskANN Build + Post-Processing (Baseline):** In this baseline, we build a regular DiskANN graph without any diversity constraints. To answer a query, we first invoke the regular DiskANN search algorithm to retrieve $r \gg k$ candidates, again without any diversity constraints. Then we iterate over the retrieved elements in sorted order of distances to the query, and greedily include the ones which do not violate the $k'$ diversity constraint, until we have $k$ total elements.

**Standard DiskANN Build + Diverse Search:** In this improvement, we use our diversity-preserving search Algorithm 5 discussed in the Appendix C, but the index construction remains the standard DiskANN algorithm.

**Diverse DiskANN Build + Diverse Search:** For our complete algorithm, we additionally use our diversity-aware index construction Algorithm 7 (Appendix C) which ensures sufficient edges are present to nodes of different colors in any neighborhood.

For all of the above algorithms, we use the parameters of list-size $L = 200$ and graph-degree 64 when building the graphs. For search, we search for $k = 100$ nearest neighbors with a diversity constraint of no more than $k' = 10$ and $k' = 1$ results per color. We vary the list size $L$ at search time to get varying quality search results and plot the recall@100[4] vs average query latency.

## 4.2 DISCUSSION

As one can see from the plot in Figure 2 (left), both of our algorithmic innovations play a crucial role in the overall search quality on the real-world dataset. For example, to achieve 95% recall@100 in the real-world dataset, the baseline approach has latencies upwards of 8ms, while the improved search algorithm brings it down to $\approx 4.5$ms. Making both build and search diverse further brings this down to around $\approx 1.5$ms, resulting in an improvement upwards of 5X.

The plot in Figure 2 (middle) reveals an interesting phenomenon: for high recalls (say 90%) on the semi-synthetic arXiv dataset, the post-processing approach has a latency of around 90ms, while the diverse search algorithm when run on the standard graph has a latency of around 135ms. This is

---

[4]Recall@100 is the size of the intersection of the algorithm's 100 returned results with the true 100 closest diverse candidates, averaged over all queries. The ground-truth set of top 100 diverse NNs for any query can be computed by iterating over all the vectors in sorted order of distances to the query, and greedily including the ones which do not violate the $k'$ diversity constraint, until we have accumulated $k$ total elements.

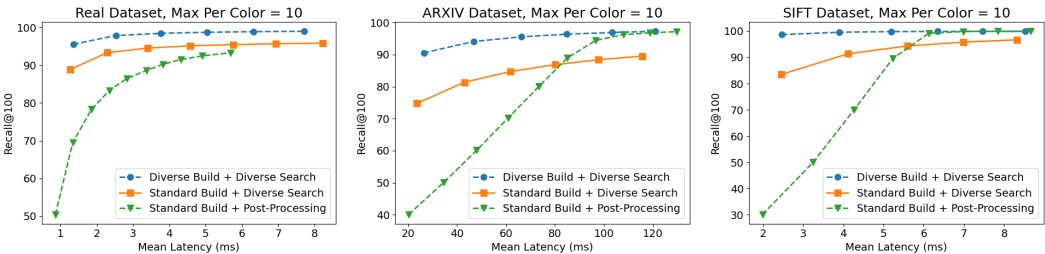

Figure 2: Recall vs Latency for real-world (left), ArXiv (middle) and SIFT (right) datasets with $k' = 10$.

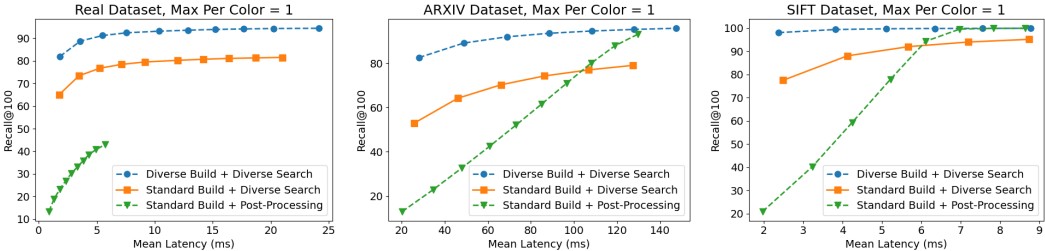

Figure 3: Recall vs Latency for real-world (left), ArXiv (middle) and SIFT (right) datasets with $k' = 1$.

perhaps because the standard graph construction might not have sufficiently many edges between nodes of different colors to ensure that the diverse search algorithm converges to a good local optimum. On the other hand, running the diverse search on the graph constructed keeping diversity in mind during index construction fares the best, with a latency of only around 25ms. A similar phenomenon occurs in the SIFT semi-synthetic dataset as well.

**Build Diversity Parameter Ablation.** In our heuristic graph construction algorithm (see Algorithms 6 and 7), the graph edges are added by considering *both the geometry of the vectors and the corresponding colors*. Loosely, the $\alpha$-pruning rule of DiskANN dictates that an edge $(u, v)$ is blocked by an existing edge $(u, w)$ if $d(w, v) \le d(u, v)/\alpha$. In the original DisKANN algorithm, any edge $(u, v)$ which is blocked is not added. In our setting, we additionally enforce that *an edge needs to be blocked by edges of $m$ different colors* to not be added to the graph, where $m$ is a tuneable parameter. We now perform an ablation capturing the role of $m$ in the graph quality using the SIFT dataset.

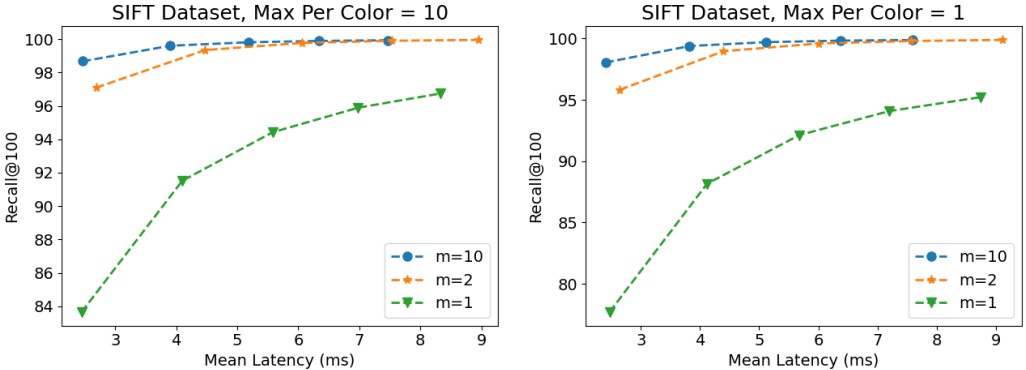

Figure 4: Recall vs Latency for SIFT dataset with $k' = 10$ (left) and $k' = 1$ (right) by varying the diversity parameter $m$ during index construction. Higher $m$ implies more diversity.

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

## A  OMITTED PROOFS FROM SECTION 3

**Lemma 3.1.** *If $P$ has a subset $\mathsf{OPT}$ of size $k$ that is $(k', C)$-diverse, our initialization algorithm finds a $(k', C/4)$-diverse subset of size $k$. (Proof in Appendix A)*

*Proof.* Note that it is straightforward to see why the set $\mathsf{SOL}$ that we get at the end is $(k', C/4)$-diverse. This is because first of all, each time we pick $k'$ points in a ball $B$ and add them to $\mathsf{SOL}$, we make sure that no additional point will ever be picked in $2B$ and thus within distance $C/4$ of the points we pick there will be at most $k'$ points in the end. Second, at the end, every remaining ball

of radius $C/4$ has less than or equal to $k'$ points in it. Therefore, we can pick all such points in the solution and everything we picked will be $(k', C/4)$ diverse.

Next we argue that we are in fact able to pick at least $k$ points in total which completes the argument. We do it by following the procedure of our algorithm and comparing it with OPT. At each iteration of the while loop that we remove $P \cap 2B$, we add exactly $k'$ points from $P \cap 2B$ to our solution SOL. Now note that the optimal solution OPT cannot have more than $k'$ points in $2B$ because by triangle inequality any pair of points in $2B$ have distance at most $C$, and picking more than $k'$ points in this ball contradicts the fact that OPT is $(k', C)$ diverse. Thus we can have an one-to-one mapping from each point in OPT $\cap 2B$ to the $k'$ points in $P \cap 2B$ added to SOL. At the end of the while iteration, we know any unmapped point in OPT still exists in $P$, so we just map it to itself. By doing this, we can have an one-to-one mapping from OPT to SOL, which means that $|\text{SOL}| \geq |\text{OPT}| = k$. □

**Proposition 3.4.** *In Line 12 of Algorithm 1, let* $\text{rep}[u]$ *be the output of greedily choosing $k/k'$ points in* $\text{bag}[u]$ *maximizing pairwise diversity. If a point $p \in \text{bag}[u] \setminus \text{rep}[u]$, we have* $\min\limits_{v \in \text{rep}[u]} \rho(p, v) \leq \min\limits_{v_1, v_2 \in \text{rep}[u]} \rho(v_1, v_2)$. *(Proof in Appendix A)*

*Proof.* For the sake of contradiction, suppose $\min\limits_{v \in \text{rep}[u]} \rho(p, v) > \min\limits_{v_1, v_2 \in \text{rep}[u]} \rho(v_1, v_2)$, and the pairwise diversity minimizer is achieved by $\min\limits_{v_1, v_2 \in \text{rep}[u]} \rho(v_1, v_2) = \rho(x, y)$. Without loss of generality, we assume $x$ is added to $\text{rep}[u]$ before $y$. At the time step $t$ when $y$ was added to $\text{rep}_t[u]$, $\min\limits_{v \in \text{rep}_t[u]} \rho(y, v) = \rho(x, y)$ and $\min\limits_{v \in \text{rep}_t[u]} \rho(p, v) \geq \min\limits_{v \in \text{rep}[u]} \rho(p, v) > \rho(x, y)$, so $y$ wouldn't have been chosen by the greedy algorithm. Therefore, we have derived a contradiction. □

# B ANALYSIS OF OTHER ALGORITHMS

## B.1 IMPROVED ANALYSIS FOR THE PRIMAL ALGORITHM WHEN $k' = 1$

In this section we give an improved analysis for the Algorithm 2 when we have $k' = 1$. First of all the algorithm has an improved number of steps and thus the runtime by a factor of $k$. Second, the solution provides an entrywise guarantee, where for each $i \leq k$ $D(q, p_i) \leq (\frac{\alpha+1}{\alpha-1} + \varepsilon)D(q, p_i^*)$

**Theorem B.1.** *Given the graph constructed by Algorithm 1, the search Algorithm 2 finds a $(1, 0.1C)$-diverse solution* ALG *satisfying* $\text{ALG} \leq \left(\frac{\alpha+1}{\alpha-1} + \epsilon\right) \cdot \text{OPT}$ *for any $(1, C)$-diverse solution* OPT *in $T = O(\log_\alpha \frac{\Delta}{\epsilon})$ steps and each step takes $\tilde{O}((8\alpha)^d k^3 \log \Delta)$ time.*

**Lemma B.2.** *Let* $\text{OPT} = \{p_1^*, ..., p_k^*\}$ *be any $(1, C)$-diverse NN solution, and* $\text{ALG} = \{p_1, ..., p_k\}$ *be any $(1, 0.1C)$-diverse NN solution. There exists another $(1, 0.2C)$-diverse NN solution* $\text{ALG}' = \{p_1', ..., p_k'\}$ *such that*

1. $p_i' \in N_{out}(p_i)$ *for any $p_i' \in \text{ALG}'$.*

2. $D(p_i', q) \leq D(p_i, q)/\alpha + \text{OPT}_i(1 + 1/\alpha)$.

*Proof.* For any point $p_i \in \text{ALG}$, we let $u$ be the point where $p_i^* \in bag[u]$ at the time we are constructing $p_i$'s our neighbors in Algorithm 1. We consider the following three cases.

Case 1: $p_i^* \in \text{rep}[u]$.

Case 2: $\min\limits_{w \in \text{rep}[u]} \rho(w, p_i^*) \leq 0.4C$

Case 3: $\min\limits_{w \in \text{rep}[u]} \rho(w, p_i^*) > 0.4C$

We construct the desired $ALG'$ in a specific order. For any index $i$ that satisfies case 1, we know $(p_i, p_i^*) \in E$, so we directly set $p_i' = p_i^*$. For any index $i$ that satisfies case 2, we set $p_i' = z \in rep[u]$ to be the point satisfying $\rho(z, p_i^*) \leq 0.4C$, which is connected to $p_i$. Next,

because all the balls $\{B_\rho(p_1^*, C/2), ..., B_\rho(p_k^*, C/2)\}$ are disjoint, the selected points up to now satisfy the $(1, 0.2C)$-diverse criteria. Then, we consider each remaining index $i$ satisfying case 3 (in any order). Let $\mathsf{rep}[u] = \{z_1, ..., z_k\}$. Because $\min_{x,y\in\mathsf{rep}[u]} \rho(x, y) > 0.4C$, their balls $\{B_\rho(z_1, 0.2C), ..., B_\rho(z_k, 0.2C)\}$ are disjoint. By a counting argument, there must exist at least one $z_j$ whose ball $B_\rho(z_j, 0.2C)$ contains no other pre-selected $p'$'s before index $i$. We then set $p_i' = z_j$, which is also connected to $p_i$. Now we get a solution $\mathsf{ALG}' = \{p_1', ..., p_k'\}$ which is $(1, 0.2C)$-diverse and each $p_i' \in N_{out}(p_i)$. To prove the distance bound, for an index $i$ satisfying case 1, $p_i' = p_i^*$, so the distance bound is valid. Otherwise we have the following:

$$
\begin{aligned}
D(p_i', q) &\le D(p_i^*, q) + D(p_i', p_i^*) \\
&\le D(p_i^*, q) + D(p_i', u) + D(u, p_i^*) \\
&\le D(p_i^*, q) + D(p_i, u)/(2\alpha) + D(p_i, u)/(2\alpha) &&\text{(Line 9 in Algorithm 2)} \\
&\le D(p_i^*, q) + D(p_i, u)/\alpha \\
&\le D(p_i^*, q) + D(p_i, p_i^*)/\alpha &&\text{($u$ is ordered earlier than $p_i^*$)} \\
&\le D(p_i^*, q) + D(p_i, q)/\alpha + D(p_i^*, q)/\alpha \\
&\le D(p_i, q)/\alpha + OPT_i(1 + 1/\alpha)
\end{aligned}
$$

$\square$

**Lemma B.3.** *Let* $\mathsf{OPT} = \{p_1^*, ..., p_k^*\}$ *be any* $(1, C)$-*diverse NN solution,* $\mathsf{ALG}^t = \{p_1^t, ..., p_k^t\}$ *be the solution found by the search Algorithm 2 on step $t$. We have the following guarantee:*

1. $\mathsf{ALG}^t$ *is* $(1, 0.1C)$-*diverse*

2. $D(p_i^t, q) \le D(p_i^{t-1}, q)/\alpha + \mathsf{OPT}_i(1 + 1/\alpha)$.

*Proof.* For the solution $\mathsf{ALG}^{t-1}$ at step $t-1$, by Lemma B.2, we know that there exists a $(1, 0.2C)$-diverse solution $\mathsf{ALG}' = \{p_1', ... p_k'\}$ where $p_i' \in N_{out}(p_i^{t-1})$. In the following, we will prove that the solution $\mathsf{ALG}^t = \{p_1^t, ..., p_k^t\}$ found by the search Algorithm 2, ordered based on increasing distance from $q$, is no worse than $\mathsf{ALG}'$ entry-wise, i.e. $D(p_i^t, q) \le D(p_i', q)$.

Let $U = \bigcup_{p_i^{t-1}\in\mathsf{ALG}^{t-1}} N_{out}(p_i^{t-1})$. We start from the solution $\mathsf{SOL}^0 = \mathsf{ALG}'$ and iterate over the set $U = \{u_1, ..., u_m\}$ in the order of increasing distance from $q$. At each iteration $i \le m$, we define $\mathsf{SOL}^i$. We will inductively show that, at the time after we consider $u_i$, the current solution $\mathsf{SOL}^i \cap \{u_1, ..., u_i\} = \mathsf{ALG}^t \cap \{u_1, ..., u_i\}$. Suppose this conclusion holds up to $i-1$ and we are considering $u_i$. We decmopose $\mathsf{SOL}^{i-1}$ into $\mathsf{PRE} = \mathsf{SOL}^{i-1} \cap \{u_1, ... u_{i-1}, u_i\}$ and $\mathsf{SUF} = \mathsf{SOL}^{i-1} \cap \{u_{i+1}, ..., u_m\}$, based on whether the point has distance to $q$ smaller or larger than $D(u_i, q)$. First, if Algorithm 2 has already added $k$ points to $\mathsf{ALG}$, we simply set $\mathsf{SOL}^i = \mathsf{SOL}^{i-1}$ from now on. Otherwise, Algorithm 2 would try to add $u_i$ to $\mathsf{ALG}^t$ if it is not conflicting with the $(1, 0.1C)$-diverse criteria. If $u_i \in \mathsf{SOL}^{i-1}$, $\mathsf{PRE} \subseteq \mathsf{SOL}^{i-1}$ is $(1, 0.1C)$-diverse, so Algorithm 2 will add $u_i$ to $\mathsf{ALG}^t$ as well. We simply set $\mathsf{SOL}^i = \mathsf{SOL}^{i-1}$ and the instruction follows. If the current $u_i \notin \mathsf{SOL}^{i-1}$, we have the following two cases:

Case 1: $\mathsf{PRE} \cup u_i$ is not $(1, 0.1C)$-diverse. In this case $u_i$ won't be added to $\mathsf{ALG}^t$, and we keep the same $\mathsf{SOL}^i = \mathsf{SOL}^{i-1}$.

Case 2: $\mathsf{PRE} \cup u_i$ is $(1, 0.1C)$-diverse. In this case $u_i$ will be added to $\mathsf{ALG}^t$. Because $\mathsf{SUF} \subseteq \mathsf{ALG}_0'$ is $(1, 0.2C)$-diverse, there exists at most one point $v \in \mathsf{SUF}$ s.t. $\rho(u_i, v) < 0.1C$. If such $v$ doesn't exist, we let $v$ to be the point with the maximal distance from $q$ in set $\mathsf{SUF}$. Then, we let $\mathsf{SOL}^i = \mathsf{SOL}^{i-1} \setminus v \cup u_i$, which is still $(1, 0.1C)$-diverse. Because $D(u_i, q) \le D(v, q)$, we have $\mathsf{SOL}^i \le \mathsf{SOL}^{i-1}$.

Therefore, in the end we have $\mathsf{SOL}^m = \mathsf{ALG}^t$ and by the transtivity property, we know $\mathsf{ALG}^t \le \mathsf{ALG}'$. Applying the distance bound for $\mathsf{ALG}'$ from Lemma B.2, we can get the same distance bound for $\mathsf{ALG}^t$. The $(1, 0.1C)$-diverse property of $\mathsf{ALG}^t$ is maintained throughout the algorithm process. $\square$

*Proof of Theorem B.1.* By Lemma B.3, we know that for any fixed $(1, C)$-diverse NN solution $\mathsf{OPT} = \{p_1^*, ..., p_k^*\}$, at each time step $t$, we can find a new $(1, 0.1C)$-diverse solution $\mathsf{ALG}^t$ and $D(p_i^t, q) \leq D(p_i^{t-1}, q)/\alpha + D(p_i^*, q)(1 + 1/\alpha)$ for any $i \in [k]$. Following the same proof argument as in Theorem 1.1 applied to every index $i$, we can get a similar entry-wise approximation bound $\mathsf{ALG}^\mathsf{T} \leq \left(\frac{\alpha+1}{\alpha-1} + \epsilon\right) \cdot \mathsf{OPT}$ in $T = O\left(\log_\alpha \frac{\Delta}{\epsilon}\right)$ steps. The $O((8\alpha)^d k^3 \log \Delta)$ time spent on each step is to get all the connected points in set $U$ and check whether each of them can be added to $\mathsf{ALG}$. $\qquad\square$

### B.2 ANALYSIS FOR THE DUAL DIVERSE NN ALGORITHM

In this section we analyze Algorithm 3.

---

**Algorithm 3** Search algorithm for dual diverse NN

---

1: **Input**: A graph $G = (V, E)$ with $N_{out}(p)$ be the out edges of $p$, query $q$, distance bound $R$, distance approximation error $\epsilon$
2: **Output**: A set of $k$ points $\mathsf{ALG}$.
3: $\mathsf{ALG} \leftarrow \{p_1, ..., p_k\}$ picked by the greedy algorithm of Gonzales for approximately maximizing the minimum pairwise diversity distance.
4: $\overline{C} \leftarrow 4 \min\limits_{p_i, p_j \in \mathsf{ALG}} \rho(p_i, p_j)$
5: **while** $\max\limits_{p \in \mathsf{ALG}} D(p, q) > (\frac{\alpha+1}{\alpha-1} + \epsilon) \cdot R$ **do**
6: $\quad \overline{C} \leftarrow \overline{C}/2$
7: $\quad$ **for** $i = 1$ to $c \cdot \log_\alpha \frac{\Delta}{\epsilon}$ **do**
8: $\quad\quad U \leftarrow \bigcup\limits_{p \in \mathsf{ALG}} (N_{out}(p) \cup p)$ and sort $U$ based on their distance from $q$
9: $\quad\quad \mathsf{ALG} \leftarrow \varnothing$
10: $\quad\quad$ **for** each point $u \in U$ in order **do**
11: $\quad\quad\quad$ **if** $\mathsf{ALG} \bigcup u$ is $(1, 0.1\overline{C})$-diverse **then**
12: $\quad\quad\quad\quad \mathsf{ALG} \leftarrow \mathsf{ALG} \cup u$
13: $\quad\quad\quad$ **if** $|\mathsf{ALG}| = k$ **then**
14: $\quad\quad\quad\quad$ Break
15: **Return** $\mathsf{ALG}$

---

**Theorem B.4.** *Given the graph constructed by Algorithm 1, the search Algorithm 3 finds a $(1, 0.05C)$-diverse NN solution $\mathsf{ALG}$ satisfying $ALG_k \leq \left(\frac{\alpha+1}{\alpha-1} + \epsilon\right) \cdot R$ in $\tilde{O}\left((8\alpha)^d k^3 \log \frac{\Delta}{\epsilon}\right)$ time, if there exists a $(1, C)$-diverse solution $\mathsf{OPT}$ with $\mathsf{OPT_k} \leq R$.*

*Proof.* For the initial solution $\mathsf{ALG} = \{p_1, ..., p_k\}$ selected by the greedy algorithm of Gonzales, we know there doesn't exist a set of $k$ points with minimum pairwise distance greater than $2 \min\limits_{p_i, p_j \in \mathsf{ALG}} \rho(p_i, p_j)$. Therefore, for the initialization $\overline{C} = 4 \min\limits_{p_i, p_j \in \mathsf{ALG}} \rho(p_i, p_j)$, we have $\overline{C}/2 \geq C$ where there exists a $(1, C)$-diverse solution $\mathsf{OPT}$ with $\mathsf{OPT_k} \leq R$.

Then our Algorithm 3 is basically adding a binary search to Algorithm 2. Invoking the analysis from Theorem B.1, if there exists a $(1, C)$-diverse solution $\mathsf{OPT} = \{p_1^*, ..., p_k^*\}$ with $\mathsf{OPT_k} \leq R$, we can find a $(1, 0.1C)$-diverse solution $\mathsf{ALG} = \{p_1, ..., p_k\}$ with $ALG_k \leq \left(\frac{\alpha+1}{\alpha-1} + \epsilon\right) \cdot R$ in $O(\log_\alpha \frac{\Delta}{\epsilon})$ steps where each step takes $\tilde{O}((8\alpha)^d k^3 \log \Delta)$ time. As a result, each time when the algorithm enters the while loop on Line 5 in Algorithm 3, we know that there doesn't exist a $(1, \overline{C})$-diverse solution with maximal distance smaller than $R$. When we exit the while loop, the current $\overline{C}$ value is at least $1/2$ of the optimal $C$ value, and the current $\mathsf{ALG}$ solution we get is at least $(1, 0.05C)$-diverse. $\quad\square$

## C ALGORITHM IMPLEMENTATION

To conduct our experiments, we provide the heuristic algorithm that we designed for the $k'$-colorful nearest neighbor problem, based on the provable algorithms provided in the main paper. The provable

indexing algorithm (1) has a runtime which is quadratic in the size of the data set and is slow in practice. This situation mimics the original DiskANN algorithm (Jayaram Subramanya et al., 2019), where the "slow preprocessing" algorithm has provable guarantees (Indyk & Xu, 2023) but quadratic running time, and was replaced by a heuristic "fast preprocessing" algorithm used in the actual implementation (Jayaram Subramanya et al., 2023). Here, Algorithm 7 offers a fast method tailored for the $k'$-colorful case, using several heuristics to improve the runtime. In the following section, we present the pseudocode for the procedures: search, index build, and the pruning procedure required for the index build.

**Diverse Search** Our diverse search procedure, is a greedy graph-based local search method. In our search method, in each step, we maintain a list of best and diverse nodes, ensuring that at most $k'$ points are selected in the list per color. In each iteration of our search algorithm, we choose the best unexplored node and examine its out neighbors. From the union of our current list and the out neighbors, we select the best diverse set of nodes while satisfying the $k'$-colorful diversity constraint—meaning no color can have more than $k'$ points in the updated list. To identify the optimal diverse set from the union, we use a priority queue designed to accommodate the diversity constraint. Below, we present the pseudocode for this diverse priority queue.

---

**Algorithm 4** Insert $(p, d, c)$ into DiversePriorityQueue (Q, $L$, $k'$)

---

1: **Input**: Current queue $Q$, tuple $(p, d, c)$ of (point, distance, color) for new insertion, maximum size $L$ of the queue, maximum size $k'$ per color.
2: **Output**: Updated queue $Q$ after inserting $(p, d, c)$ which maintains the best set of at most $L$ points and at most $k'$ points of each color.
3: Let count$(c) \leftarrow$ number of elements in $Q$ with the color $c$.
4: Let maxDist$(c) \leftarrow$ maximum distance of element in $Q$ with color $c$.
5: **if** count$(c) \leq k'$ **or** $d < $ maxDist$(c)$ **then**
6:     Insert $(p, d, c)$ into $Q$
7:     **if** count$(c) > k'$ **then**
8:         Remove the element with the maximum distance in $Q$ having color $c$.
9: **if** $|Q| > L$ **then**
10:     Remove the element with the maximum distance in $Q$.

---

Building on the previous explanation of the diverse priority queue, we outline the description of our diverse search procedure as follows.

---

**Algorithm 5** DiverseSearch$(G, s, q, k', k, L)$

---

1: **Input**: A directed graph $G$, start node $s$, query $q$, max per color parameter $k'$, search list size $L$.
2: **Output**: A set of $k$ points such that at most $k'$ points from any color.
3: Initialize DiversePriorityQueue $\mathcal{L} \leftarrow \{(s, D(s, q), col[s])\}$ with color parameter $k'$ and size parameter $L$.
4: Initialize a set of expanded nodes $\mathcal{V} \leftarrow \emptyset$
5: **while** $\mathcal{L} \setminus \mathcal{V} \neq \emptyset$ **do**
6:     Let $p^* \leftarrow \underset{p \in \mathcal{L} \setminus \mathcal{V}}{\arg\min} D(p, q)$
7:     $\mathcal{V} \leftarrow \mathcal{V} \cup \{p^*\}$
8:     Insert $\{(p, D(p, q), col[p]) : p \in N_{\text{out}}(p^*)\}$ to $\mathcal{L}$
9: **Return** [top $k$ NNs from $\mathcal{L}; \mathcal{V}$]

---

**Diverse Prune:** A key subroutine in our index-building algorithm is the prune procedure. Given a node $p$ and a set of potential outgoing edges $\mathcal{V}$, the standard prune procedure removes an edge to a vertex $w$ if there exists a vertex $u$ such that an edge $p \rightarrow u$ exists and the condition $D(u, w) \leq \frac{D(p, w)}{\alpha}$ is satisfied. Intuitively, this means that to reach $w$, we would first reach $u$, thus making multiplicative progress and eliminating the need for the edge $p \rightarrow w$, which contributes to the sparsity of the graph.

However, to account for diversity, the outgoing edges from the node must also be diverse and enable access to multiple colors. To address this requirement, we modify the standard prune procedure to incorporate the diversity constraint. The details of our revised algorithm are provided next.

---

**Algorithm 6** DiversePrune$(p, \mathcal{V}, \alpha, R, m)$

---

1: **Input**: A point $p$, set $\mathcal{V}$, prune parameter $\alpha$, degree parameter $R$ and diversity parameter $m$.
2: **Output**: A subset $\mathcal{V}' \subseteq \mathcal{V}$ of cardinality at most $R$ to which edges are added.
3: Sort all points $u \in \mathcal{V}$ based on their distances from $p$ and add them to list $\mathcal{L}$ in that order.
4: Initialize sets blockers$[u] \leftarrow \emptyset$ for each $u \in \mathcal{V}$.
5: **while** $\mathcal{L}$ is not empty **do**
6: $\quad u \leftarrow \underset{u \in \mathcal{L}}{\operatorname{argmin}} D(u, p)$
7: $\quad \mathcal{V}' \leftarrow \mathcal{V}' \cup \{u\}$ and $\mathcal{L} \leftarrow \mathcal{L} \setminus \{u\}$
8: $\quad$ **if** $|\mathcal{V}'| = R$ **then break**
9: $\quad$ **for** each point $w \in \mathcal{L}$ **do**
10: $\quad\quad$ **if** $D(u, w) \leq D(p, w)/\alpha$ **then**
11: $\quad\quad\quad$ blockers$[w] \leftarrow$ blockers$[w] \cup \{col(u)\}$
12: $\quad\quad\quad$ **if** $|$blockers$[w]| = m$ or $col(u) = col(w)$ **then**
13: $\quad\quad\quad\quad \mathcal{L} \leftarrow \mathcal{L} \setminus \{w\}$
14: **Return** $\mathcal{V}'$

---

**Diverse Index:** Our indexing algorithm follows the same approach as the DiskANN "fast prepro­cessing" heuristic implementation (Jayaram Subramanya et al., 2023), but we replace the search and prune procedures in their implementation with our diverse search and diverse prune procedures. The details of our index-building procedure are provided below.

---

**Algorithm 7** DiverseIndex$(P, \alpha, L, R, m)$

---

1: **Input**: A set of $n$ points $P = \{p_1, \ldots, p_n\}$, prune parameter $\alpha$, search list size $L$, degree parameter $R$ and diversity parameter $m$.
2: **Output**: A directed graph $G$ over $P$ with out-degree at most $R$.
3: Let $s$ denote the estimated medoid of $P$.
4: Initialize $G$ with start node $s$.
5: **for** each $p_i \in P$ **do**
6: $\quad$ Let $[\mathcal{L}; \mathcal{V}] \leftarrow$ DiverseSearch $(G, s, p_i, k' = L/m, L, L)$
7: $\quad$ Let $\mathcal{V}' =$ DiversePrune $(p_i, \mathcal{V}, \alpha, R, m)$.
8: $\quad$ Add node $p_i$ to $G$ and set $N_{\text{out}}(p_i) = \mathcal{V}'$ (out-going edges from $p_i$ to $\mathcal{V}'$).
9: $\quad$ **for** $p \in N_{out}(p_i)$ **do**
10: $\quad\quad$ Update $N_{\text{out}}(p) \leftarrow N_{\text{out}}(p) \cup \{p_i\}$.
11: $\quad\quad$ **if** $|N_{\text{out}}(p)| > R$ **then**
12: $\quad\quad\quad$ **Run** DiversePrune$(p, N_{\text{out}}(p), \alpha, R, m)$ to update out-neighbors of $p$.

---

