# OpenReview forum: "Diverse Graph-based Nearest Neighbor Search"
_ICLR.cc/2025/Conference — Submitted to ICLR 2025_

### Official Review · Reviewer_aySC · 2024-10-27

**Soundness:** 2
**Presentation:** 1
**Contribution:** 3
**Rating:** 5
**Confidence:** 3

**Summary:**

The paper studies the diversity-aware ANNS given a query q, a data set P, a distance measure D(), a diversity measure \rho().
Diversity-ANNS asks to report an approximate kNN S, |S| = k, that contains points closest to q given D(), and maximizes the diversity measurement \rho() which is the minimum value of \rho(p, p') for any pair p, p' \in S.

The work proposes two different variants of diversity-aware ANNS, including
(1) Primal version: Given a fixed diversity threshold C, guarantee to find a set (C/a)-diversity S where the kNN_dist(q, S) \leq kNN_dist(q, S*), where S* is the optimal solution.
There are two approximation factors here, including $a$ for diversity measure \rho(), and $c$ for distance measure D().
(2) Dual version: Given a fixed radius R, guarantee to find a set (C/a)-diversity S where kNN_dist(q, S) \leq cR and C is the maximum value among all subsets of size k of the ball B(q, R).

Indeed, there is an additional parameter k' that governs the maximum number of points in the returned kNN set S that violates the diversity constraint for each point in S.
The dual version has been studied with LSH-based solution (Abbar et al. 2013b) and the primal version is resolved in this work using graph-based index.

The main contribution is that the proposed solution for the primal version avoiding the approximation factor c is derived from from the graph-based index (Indyk & Xu 23, DiskANN).
In the DiskANN indexing phase, the work adds one step on top of the graph-based index with diversity-aware algorithm (via the Gonzalets greedy algorithm that maximizes the minimum pairwise \rho() distance) to prune candidates in the list L (Line 12 - Alg 1).
In the querying phase, there is an initialization step that picks the diversity-aware seeds (Line 3 - Alg 2) together their neighbors as candidate points. The greedy search over the graph index iteratively keeps finding the kNN set S of the query q and diversifying S given \rho().

The main theoretical results (stated in Theorem 1.1) show how to guaratee the answer for the primal version with  a constant a, c = (\alpha + 1) \ (\alpha - 1) where \alpha is the parameter of the DiskANN index.
The query time complexity is independent on n and exponentially dependent on the doubling dimension d.
The indexing space complexity is linear in n and exponential in d.
Both graph-based querying time and indexing space are better than LSH-based approaches, in terms of the dependency on n.
Some empirical results on real-world data sets demonstrate the efficiency of the proposed indexing and querying solutions compared to standard graph-based index baselines (DiskANN + postprocess, DiskANN + diverse query) given the recall-latency tradeoff.

**Strengths:**

S1. The diversity-award ANNS is important in many recommendation applications, and the paper is the first work with graph-based index solution (to the best of my knowledge)

S2. The solutions comes with theoretical backing on the approximation factors in terms of diversity and distance measures.

S3. The graph-based indexes are versatile for several distance measures, so the proposed solution can piggy-back on industry released graph-based libraries on many distance measure D().

**Weaknesses:**

W1. I found the initialization step (Line 3 - Alg2) in the query phase will be the bottleneck since it might execute O(n) range queries given the \rho distance in the worst case. This step will dominate the query phase. However, there is no discussion on the complexity of this step. Since the initialization step requires C as input, so different values of C require to run the initialization step again, which is more troublesome.

W2. Though graph-based indexes show advantages compared to hash-based solutions on standard ANNS, it is not well-understood that graph-based solutions will be better than hash-based solutions on diversity-awared ANNS, especially given additional steps on querying phase over the graph-based indexes. The experiments do not contain hash-based baselines, which does not convince the advantage of the proposed solution given the diversity constraints.

W3. While the proposed theory (Theorem 1.1) works on a general \rho, the empirical results consider the specific \rho ('colorful' version), which clearly not align well to the theory. As stated in Line 152, there is no role of \rho in the practical implementation, which really weakens the contribution of theory.

W4. The emprical setting is not clearly described. For example, since standard greedy search on graph will be much faster than diverse search, one can set r much larger than k to expect get better recall with post-processing. Also, since we consider k = 100, I wonder why we only set L = 200 since this value seems to be the default settting of graph-based index for k = 10.

W5. The paper is very difficult to follow. Section 3 describes the indexing and quering algorithms without any intuitions to show why the author proposed modifications from DiskANN. While the paper builds on graph-based indexes, it is difficult the see the key differences in the manuscript, and it is hard to see the novelty in both indexing and querying phases. There are nearly 3 pages dedicating for the proof. It might be better to move the proof to the appendix to have room to explain the algorithms in high levels.


Minor things.

Line 125: OPT_k has not defined before. Is it the optimal kNN distance among all possible answered set for the value C?

Line 248: Lemma 3.1 ?

**Questions:**

Please address the weaknesses W1 - W4, and the following questions.

Q1. Could you provide a complexity analysis of the initialization step and discuss how it impacts the overall query time. How does your algorithm's performance change for different values of C, given that the initialization needs to be rerun?

Q2. Could you add hash-based solutions baselines for diversity-aware ANNS? This would provide a more comprehensive evaluation of the proposed method's advantages.

Q3. What is the value of r, and the fraction of running time of the post-processing step in the query time?

Q4. How sensitive are the emprical results when k = {10, 20} with L = 200, or k = 50, L = 500?

---

> ### Author Response · Authors · 2024-11-20
> **Author Response to Reviewer aySC -- 1**
>
> We thank the reviewer for helpful feedback. Below we address specific comments and questions.
>
> >*“The dual version has been studied with LSH-based solution (Abbar et al. 2013b) and the primal version is resolved in this work using graph-based index”*
>
> We would like to clarify that, in this paper,  we give graph-based algorithms for *both* the primal and the dual version of the problem, whereas in prior work of Abbar et al that used LSH, only the dual problem was solved.
>
> >*“Could you provide a complexity analysis of the initialization step and discuss how it impacts the overall query time.”*
>
> To clarify,  the initialization does *not* depend on the query and thus needs to be run only once over all the queries. It is true that different values of $C$ require different initialization steps but only $\log \Delta$ such searches suffice.
> The runtime of the initialization algorithm is indeed $O(n^2)$. However, we present this initialization algorithm mostly as a backup option to handle very hard instances which we do not expect to occur in practice. In particular, in our experiments we simply sample random points and verify they are sufficiently diverse.
>
> >*“Though graph-based indexes show advantages compared to hash-based solutions on standard ANNS, it is not well-understood that graph-based solutions will be better than hash-based solutions on diversity-awared ANNS. ... Could you add hash-based solutions baselines for diversity-aware ANNS?”*
>
> Because of their wide applicability of graph-based methods, the focus of this work is to provide diversity aware search that specifically uses the graph-based methods. Since many companies already use graph-based methods, switching to other methods is less preferable. For example, one product group may want to use a single (standard) indexing algorithm and depending on the query, either use standard search or diverse search.
>
> That said, we also reached out to the authors of the diverse NNS using LSH-based methods and we were informed that they unfortunately no longer have the code or datasets, as those papers were written over one decade ago.
>
> >*“While the proposed theory (Theorem 1.1) works on a general \rho, the empirical results consider the specific $\rho$ ('colorful' version), which clearly not align well to the theory.”*
>
> This special case of $\rho$ was motivated by the application where the goal was to show $k$ product ads to the user such that they all come from different sellers. It turned out that  a more relaxed version, in which we require the number of ads from a single seller be bounded by at most $k’$, also works well for the user perspective. Thus, our experimental evaluation is focused on this case.
>
> We note that the diversity metric \rho used in Abbar et al is different from ours - it is induced by the distance in one dimension. This demonstrates practical use cases that go beyond the colorful case.  However, their data sets are unfortunately not available.
>
> >*“The emprical setting is not clearly described. For example, since standard greedy search on graph will be much faster than diverse search, one can set r much larger than k to expect get better recall with post-processing. Also, since we consider k = 100, I wonder why we only set L = 200 since this value seems to be the default setting of graph-based index for k = 10.”*
>
> For the post-processing approach, we precisely do what you outline. We seek $r \gg k$ candidates using a vanilla search, and then run a diversification step as post-processing. We vary $r$ to as high as 1000 in this step. For instance: if we use the non-diverse build and do the standard search with a post-processing step (Baseline), to achieve >90% recall for Max per color = 10, the values of $r$ are roughly 800, 700 and 400 for Real, Arxiv and SIFT datasets respectively. For Max per color =1, the values of $r$ are roughly 1000 and 500 for Arxiv and SIFT datasets. However on the Real data set the baseline algorithm doesn’t even achieve 50% recall even for $r = 1000$. Apologies if this is not clear.
>
> The $L$ parameter is what we use for indexing. It is pretty robust in the real world to the target value of $k$.
>
> It would be great if you could point us to where $L = 200$ is the default setting for $k = 10$ in practical applications. The graph degree, and $L$ parameter are indeed tunable, but much more robust in our opinion than the parameters in other ANN approaches, across dataset dimension and recall target $k$.  For example, we ran the experiments for the SIFT dataset (Max per color=1), using $L = 400$ during the index build, and the results were very similar to that in Figure 3 in the paper that was computed using $L = 200$.
>
> >*“It might be better to move the proof to the appendix to have room to explain the algorithms in high levels.”*
>
> Thank you for your suggestion, we will move the proofs to the appendix and provide a high level overview of the algorithms and give further intuitions instead.

---

> > ### Author Response · Authors · 2024-11-20
> > **Author Response to Reviewer aySC -- 2**
> >
> > >*“Line 125: OPT_k has not defined before. Is it the optimal kNN distance among all possible answered set for the value C?”*
> >
> > Apologies if the writing was unclear. We will fix the $OPT_k$ issue (we defined $S(q)_k$ but did not extend this definition to $OPT_k$).
> >
> > >*“Line 248: Lemma 3.1 ?”*
> >
> > Lemma 3.1 proves that a valid $(k',C/4)$-diverse subset can be obtained by our initialization algorithm. Such a valid initial solution is used in line 12 of Algorithm 2 (“...initialization step proved in Lemma 3.1,” sorry for the typo). We are also happy to address any further questions, should there be any.
> >
> > >*“What is the value of r, and the fraction of running time of the post-processing step in the query time?”*
> >
> > The value of $r$ for the baseline algorithm varies at search time. Depending on how many candidates we retrieve, and pass down for post-processing, our recall improves. The value of $r$ in our experiments ranges from $100$ to $1000$. For instance: as discussed earlier, if we use the non-diverse build and do the standard search with a post-processing step (Baseline), to achieve >90% recall for Max per color$ = 10$, the values of $r$ are roughly $800$, $700$ and $400$ for Real, Arxiv and SIFT datasets respectively. For Max per color$ = 1$, the values of $r$ are roughly $1000$ and $500$ for Arxiv and SIFT datasets. However on the Real data set the baseline algorithm doesn’t even achieve 50% recall even for $r = 1000$.
> >
> > Furthermore, the post-processing step contributes to a small amount of 2% of the total running search time for the baseline algorithm.
> >
> > >*“How sensitive are the empirical results when k = {10, 20} with L = 200, or k = 50, L = 500?”*
> >
> > As suggested, to test the sensitivity of our empirical results, we performed experiments for k=20 and chose $L = 200$ during the index build. We observe a similar pattern to that of $k = 100$ showing the superiority of our approach compared to the baseline.  For example for the SIFT dataset, Max per color$ = 1$, at latency 1ms our algorithm achieves  Recall@20 of over 92%, whereas the baseline algorithm achieves only 35% recall. At latency 2ms, our algorithm achieves 98% recall whereas the baseline is at 92%.

---

> ### Comment · Reviewer_aySC · 2024-11-22
> **Still some important issues**
>
> Thank you for your detailed feedback. However, I believe my concerns remain valid and significant
>
> The paper proposes diverse-aware indexing and querying mechanisms for graph-based approximate nearest neighbor (ANN) search, where time complexity depends exponentially on the doubling dimension. The problem focuses on maximizing the diversity measure defined as the minimum value of \rho(p, p') for any pair p, p' in the returned set S.
>
> From a theoretical perspective, the contribution is difficult to compare with standard locality-sensitive hashing (LSH), where complexity depends on the original data dimension. Additionally, the proposed indexing construction time of O(n^2) for the worst-case dataset is less favorable compared to LSH.
>
> The practical setting of the paper considers the *binary* version \rho() \in {0, 1}. This relaxation reduces the theoretical significance for several reasons:
>
> - Simpler solutions exist for the binary variant e.g. building different ANN index for different color, for each color getting a set of nearest neighbors, then forming the diverse-aware solutions based on these nearest neighbors. If the number of colors (e.g. sellers) is not large, or we merge several colors with small of number of points together, then several heuristic methods can outperform your proposed solution.
>
> - The theory does not stand well on the *binary* relaxation of \rho() \in {0, 1} and C \in {0, 1}. For example, the initialization step (Line 262) might return NO solution with C = 1. I am not sure that Lemma 3.1 still holds in case C \in {0, 1}.
>
> Regarding empirical results and settings, without comparing LSH solution with \rho \in [0, 1] is a big issue to me.
>
> - Your proposed solution requires a new diverse-aware graph-based index (Alg 1), which means that we have to maintain two different graph-based indexes. The cost of building (diverse-aware) graph-based index is clearly larger than the cost of building (diverse-aware) LSH.
>
> - On semi-synthetic data sets, I wonder why not considering any applications where \rho() \in [0, 1] (e.g. LDA to extract topics) so that we will have a fair comparison with Abbar et al. The setting that use embeddings and the binary variant of \rho() is not adequate to show the advantage of your work compared to Abbar et al. which has \rho() \in [0, 1]
>
> Given the misalignment between theoretical analysis and practical settings, and the lack of comparison with LSH baselines on \rho \in [0, 1], I maintain my score.
>
> Minor things:
> - https://github.com/nmslib/hnswlib use ef_construction = 200 for k = 10.
> - Heuristic method to solve the binary version: Given that the post-processing is around 2%, consider the color from {1, 2, 3} with prob 0.9 and {4, 5, ..., 1000} with prob 0.1, we can build 3 different indexes for color 1, 2 and 3. For the rest of data, breaking it into 100 clusters and build each index for each cluster. I think such solution can solve the color version for k = 100.

---

> > ### Author Response · Authors · 2024-11-23
> > **Author response to the comment by reviewer aySC**
> >
> > Thank you for participating in the discussion and providing additional feedback.
> >
> > > Regarding the exponential dependency on doubling dimension:
> >
> > * The claim of this paper is to incorporate diversity into the graph-based NNS algorithm of diskANN. Therefore, given that the exponential dependency of diskANN on doubling dimension is unavoidable, we cannot perform any better.
> >
> > > Regarding the runtime of the initialization:
> >
> >   * Note that as an example, for our main application of Seller/colorful diversity, the initialization step translates to an easy task of just finding $k$ data points (irrespective of the query) that all have different colors (for $k’=1$). For any $1\leq k'\leq k$, this task can easily be done in $O(n)$ at the indexing time.
> >
> > > About your proposed solution of constructing a different ANN data structure per seller:
> >
> > * This is not feasible as the number of colors in our real-world application is ~2000. On the other hand, if one merges the colors with fewer points in the same bucket, in theory you can hit the same issue with that particular bucket, e.g. consider the case where the points around the query are all from those low appearing colors in the same bucket. Instead, in this work we show a theoretically provable algorithm that leads to heuristics which as we show perform well on our application.
> >
> > > Correctness of lemma 3.1 for the binary case:
> >
> > * We emphasize that lemma 3.1 holds for any metric space including the binary case. To see this, set $C=½$. Then, any $k’$-colorful solution OPT of $k$ points is $(k',C)$-diverse. Thus our algorithm finds a set of $k$ points that is $(k’,⅛)$ diverse. But because the metric is binary, this is in fact a $k'$-colorful solution. However, as mentioned above, for the binary case, the initialization becomes very simple. One just needs to find $k$ points s.t. no color appears more than $k'$ times. This is a simple task that can be done in $O(n)$. Apologies if this was not clear. If the reviewer finds it helpful, we will include the theoretical algorithm for the special case of binary diversity $\rho$.
> >
> > > Regarding comparison to diverse-aware LSH:
> >
> > * Per your suggestion, we reached out to the authors of (diverse-aware) LSH and their code is unfortunately no longer available. Moreover, they solve the dual version of the problem (given a radius $r$, they find the $k$ most diverse points within that radius), whereas our main application is on the primal version (our goal is to find the closest $k$ points to the query that satisfy a given diversity requirement). Finally, there is a real need in coming up with a diverse variant of the “graph-based” search algorithm, given it is widely used in practice.
> >
> > > About your concern on misalignment between the theory and practice:
> >
> > * We believe that our theoretical and practical contribution in this work align well. We started our work with the real-world application of seller diversity, and gave a theoretical algorithm for it which lead to heuristics that perform well on the real data we have. On the theoretical side, we further generalized our algorithm and analysis to work for arbitrary metric $\rho$.

---

### Official Review · Reviewer_KNJ1 · 2024-11-03

**Soundness:** 3
**Presentation:** 3
**Contribution:** 2
**Rating:** 6
**Confidence:** 4

**Summary:**

The paper studies approximate nearest neighbor search with diversity constraints, where instead of just minimizing the dissimilarity from the query to a set of $k$ output vectors, we also want to maximize the pairwise dissimilarity $\rho$ between the output vectors. The paper presents provably efficient graph-based algorithms for both the standard setting and the radius search setting where the results lie approximately within a pre-defined radius $R$.

Inspired by their theoretical algorithms, for practical applications, the authors propose heuristic approximations for an algorithm in the special case where each point is associated with a "color" and no more than $k'$ results can have the same color. In the experiments, this method is superior to the baseline method (querying $rk$ nearest neighbors for some $r$ and then applying post-processing) for datasets with a very skewed color distribution.

**Strengths:**

- The theoretical results are to the best of my knowledge the first for graph-based methods for the diverse nearest neighbor problem and the obtained bounds are meaningful. The analyzed algorithms are graph-based methods that are currently viewed as state-of-the-art for approximate nearest neighbor search.
- The authors also propose a practical heuristic method based on DiskANN for the diverse nearest neighbor problem. The proposed method achieves a significant improvement over vanilla DiskANN on the real-world dataset used by the authors.
- The paper is well-written and the presented arguments are easy to follow.

**Weaknesses:**

Instead of studying the performance of a practical diverse nearest neighbor search algorithm, the authors design a provably efficient theoretical algorithm. The analysis is quite similar in parts to the earlier work of Indyk & Xu (2023) in studying graph-based ANN methods. While the diverse nearest neighbors problem is relevant in many use cases, I am not convinced that the theory here is particularly insightful enough to cover for the slightly lesser importance of the problem (given the relative lack of empirical work for the problem).

The authors mention that their heuristic algorithm is "based on the provable algorithms provided in the main paper". However, I do not see how the proposed heuristics in Algorithms 5-7 are based on the theoretical version presented in Algorithms 1 and 2. It would be helpful for the authors to explicitly clarify this connection. Instead, the proposed empirical algorithm seems to be a relatively straightforward modification of the DiskANN algorithm with diversity constraints included in the search and prune procedures. Moreover, the proposed heuristic algorithm is only applicable to the "colorful" special case, while the standard post-processing approach is of course applicable to any case.

Two of the three datasets used in the experiments use semi-synthetic data where the color distributions are perhaps unrealistically skewed, causing $r$ to be much higher for the baseline than it would typically need to be. For example, for the SIFT dataset the authors sample a dominant color with probability 0.8 and the the rest of the 999 uniformly (in this case, I would just build two indexes). Moreover, you would expect there to be some correlation between the distances and the colors of given points.

**Questions:**

- [Q1] Referring (line 146) to the difference between the main bound in the paper (Theorem 1.1) and the bound presented by Indyk & Xu (2023), you say that the latter does not depend on $k$ or $k'$ as these parameters do not exist in the standard NN formulation. Of course, $k$ exists in the standard $k$-NN formulation -- how do the bounds you refer to look for $k$-NN (at a glance, the paper by Indyk & Xu seems to state a bound for 1-NN)? Do they also have a $k^3$ term for the time complexity of each step?
- [Q2] Your results depend on the aspect ratio $\Delta$ which can grow arbitrarily high when the minimum distance between any two points gets closer to zero. However, with graph methods points that are very near each other should be easier for the algorithm. Is there an intuitive explanation for this discrepancy?
- [Q3] Roughly how much does the indexing time of your practical diverse indexing method differ from that of DiskANN?
- [Q4] What value of $r$ do you use for the baseline method in the experiments and is it the same for each dataset? How high does $r$ need to be with respect to $k$ to achieve a high recall on the different datasets?
- [Q5] Could you verify that the latencies you provide in Figures 2 and 3 are correct? They seem quite high given the experimental setup if you use 48 threads per run (e.g., I would expect the queries to take << 1 ms on SIFT).
- [Q6] Is 20 the total number of sellers in the real-world data set or are there more? Would it be possible to publish the data set?
- [Q7] On the semi-synthetic data sets, I would also expect vanilla DiskANN to perform relatively much worse when $k'$ = 1 compared to when $k'$ = 10. Can you hypothesize why that is not the case?
- [Q8] What is the rationale for choosing the distributions for the semi-synthetic datasets?

---

> ### Author Response · Authors · 2024-11-20
> **Author Response to Reviewer KNJ1 -- 1**
>
> We thank the reviewer for the helpful comments. Below we address specific comments and questions.
>
> > *“The analysis is quite similar in parts to the earlier work of Indyk & Xu (2023) in studying graph-based ANN methods.”*
>
> While the overall structure of the analysis is similar to that of Indyk&Xu, the algorithms and the analysis build significantly on top of that. Even solving the problem for the much simpler case of k'=1 and binary metric $\rho$ turned out not to be easy to solve and analyze. In particular, coming up with the “right” notion of an iterative step in the search algorithm (lines 10…14) and quantifying the progress that such a step makes (Lemmas 3.3 and 3.5) took a fair amount of effort. Additional effort was needed to generalize the algorithm to arbitrary diversity metrics $\rho$ and $k’>1$.
>
> >*“However, I do not see how the proposed heuristics in Algorithms 5-7 are based on the theoretical version presented in Algorithms 1 and 2. … Instead, the proposed empirical algorithm seems to be a relatively straightforward modification of the DiskANN algorithm ...”*
>
> Our heuristic algorithm is indeed a simple generalization of DiskANN with diversity in graph build and graph search and prune. We consider this minimal code modification to be a *positive* aspect of our result because it makes it easier for the diverse algorithm to be actually implemented for various use cases.
>
> To better illustrate the connection between the theoretical and heuristic algorithms, below we present  a simplified variant of our theoretical algorithm that works only for the colorful case, for the simple case of $k’=1$.
>
> - In the indexing algorithm, in order to specify the out-neighbors of a point $p$,
>   1) We sort and keep all the other points in a list $L$, and as long as $L$ is not empty we take the closest point $u$ to $p$, and connect $p$ to $u$.
>   2) We then remove all points $v$ from $L$ such that $v$ is close to $u$ and either
>     - color[$v$] = color[$u$]
>     - Or we have already connected $p$ to $k$ points of different colors in close neighborhood of $v$.
>
> - In the search algorithm,
>   1) We start with $k$ vectors $a_1,...,a_k$ that all have different colors.
>   2) As long as there exists a point $a_i$ that has a neighbor $v$ such that
>      - $v$ is closer to the query and
>      - $v$ has a different color than all other $a_j$;
> we replace $a_i$ with the closest such neighbor to the query.
>
> We hope that the above pseudocode emphasizes the similarities between our theoretical and practical algorithms. If the reviewer agrees, we will add this description to the paper. (We did not include it earlier to avoid repetition).
>
> > *“What is the rationale for choosing the distributions for the semi-synthetic datasets?”*
>
> We believe that it is indeed important to consider such skewed distributions. For example, in the real-world dataset, the top six colors account for nearly 90% of all vectors, which is a much more top-heavy setting than our synthetic distribution. Also, while one could take the two-index approach you suggested for our highly specialized distribution, we think it is hard to generalize this to other distributions, including our real-world dataset.
>
> > *“Referring (line 146) to the difference between the main bound in the paper (Theorem 1.1) and the bound presented by Indyk & Xu (2023), you say that the latter does not depend on k or k’ as these parameters do not exist in the standard NN formulation.”*
>
> For the regular approximate nearest neighbor search problem formulation, the theoretical results typically only guarantee an approximation ratio for the top-1 returned neighbor. In particular, Theorem 3.4 in Indyk-Xu'23 indeed considers only the case of $k=1$. Unfortunately, we do not know how a generalization to $k>1$ would look like.
>
> > *“Your results depend on the aspect ratio $\Delta$ which can grow arbitrarily high when the minimum distance between any two points gets closer to $0$”*
>
> Indyk-Xu'23 show that log Delta dependence is unavoidable for DiskANN-based algorithms, even for one-dimensional point sets. Please see Theorem 3.5 and its proof in the aforementioned paper for the reasoning.
>
> > *“Roughly how much does the indexing time of your practical diverse indexing method differ from that of DiskANN?”*
>
> As discussed in the experiments section (4.2, Build Diversity Parameter Ablation), we have a tunable parameter m which enforces that an edge needs to be blocked by edges of $m$ different colors to not be added to the graph; this parameter is a proxy for $k/k'$ value which is equal to the number of different color edges that needed to be added in the ball. Higher value of m means that we are building the graph to enable higher diversity. In our experiments, higher values of $m$ yield higher build time. In the results reported in the paper, we choose $m=3$ for which the index build time is at most 2 times higher than the non-diverse build time. For $m=10$, the index build time is at most $2.6$ times higher.

---

> ### Author Response · Authors · 2024-11-20
> **Author Response to Reviewer KNJ1 -- 2**
>
> >*“What value of r do you use for the baseline method in the experiments and is it the same for each dataset?”*
>
> As per Figures 2 and 3, for the baseline method, the crucial parameter we vary at search time is the value of $r$, how many candidates to retrieve, before a diversity-aware post-processing. Higher $r$ gives higher recall. In all these experiments, the value of $r$ ranges between $100$ and $1000$. The value of $r$ depends not just on the value of $k$, but also on how non-diverse the local ball around the query is.
>
> >*“Could you verify that the latencies you provide in Figures 2 and 3 are correct?”*
>
> We reran the experiments on the SIFT dataset with Max Per Color =1 and we observed similar latencies as reported in Figure 3. There are multiple reasons why the latency numbers don’t meet your expectation: Firstly, since we are using all 48 threads for search, this will maximize throughput (queries per second) while hurting latency (which is the time taken per query), due to resource contention between various threads. In case there is ambiguity, we would like to stress that mean latency is not 1/QPS, but rather 48/QPS, so more threads would not directly reduce latency. We re-ran the experiments with 1 thread and notice a 4-5X gain in latency across the board for all experiments. However, we prefer reporting numbers with higher thread utilization because real-world scenarios indeed involve multiple threads searching the index simultaneously. Another reason for higher latencies is that, to output a diverse set of results, one needs to fetch a large number of items. Finally, a small contribution (2%) to the increase comes from the additional post-processing step for the baseline algorithm.
>
> >*“Is 20 the total number of sellers in the real-world data set or are there more?”*
>
> There are more. In the pie chart we are only showing the distribution of the top 20 sellers. The total number of sellers is approximately 2K.
>
> > *“On the semi-synthetic data sets, I would also expect vanilla DiskANN to perform relatively much worse when k’= 1 compared to when k’= 10. Can you hypothesize why that is not the case?”*
>
> It is indeed worse with $k’=1$ compared to $k’=10$ for both the Arxiv and SIFT semi-synthetic datasets. For example, for the Arxiv dataset, the recall for $k’=10$ is around 90% for a latency of 60ms, while it is only 60% for the $k’=1$ case with the same latency bound. A similar trend is seen for the SIFT dataset also. Perhaps the real-world distribution is even more skewed (6 sellers account for 90% points, so seeking 100 diverse candidates for $k’=1$ is much harder than 10 diverse results for $k’=10$), so that shows in the results. In semi-synthetic, this measure is only 80% dominated by the big sellers.

---

> > ### Comment · Reviewer_KNJ1 · 2024-11-23
> >
> > Thank you for your detailed answers to my questions. I have raised my score slightly in response.
> >
> > > To better illustrate the connection between the theoretical and heuristic algorithms, below we present a simplified variant of our theoretical algorithm that works only for the colorful case, for the simple case of $k' = 1$.
> >
> > I appreciate the authors clarifying the connection and hope that they include this and any further clarifications as they see fit in future revisions since the connection seems tenuous in the paper. It could be helpful to, e.g., move some of the proofs to the Appendix to make space for an explanation of the practical algorithm and its connection to the theory.
> >
> > > We believe that it is indeed important to consider such skewed distributions. For example, in the real-world dataset, the top six colors account for nearly 90% of all vectors, which is a much more top-heavy setting than our synthetic distribution. Also, while one could take the two-index approach you suggested for our highly specialized distribution, we think it is hard to generalize this to other distributions, including our real-world dataset.
> >
> > My problem here is that you have only one private real-world dataset, and then, inspired by this dataset, you introduce similar synthetic datasets that are also very skewed. From my experience in working with real-world vector search deployments, I do not think these skewed distributions are particularly representative. The simple baseline method works well in most cases, which explains the lack of empirical work for the problem (in contrast to many other variants of ANN search such as filtered or OOD settings).
> >
> > Finally, I also want to add that I disagree with my fellow reviewers in that I do not think a comparison to the prior LSH-based method of Abbar et al. would be meaningful, as they only consider the dual version with Hamming distance, and even if their code were available, it is highly unlikely the performance would come close to even the (modern, highly-optimized) baseline methods.

---

> ### Author Response · Authors · 2024-11-25
>
> Thank you for providing additional feedback and also for increasing your score.
>
> According to the targeted application in shopping and search (see lines 417-421), we argue that real-world datasets are more likely to be skewed because similar products tend to be sold by the same seller, and each seller’s products cover a neighborhood within the product space. Therefore, it is reasonable to expect that the distributions are skewed, as supported by our real datasets. We will consider additional experiments on real-world datasets in future versions of our paper.
>
> Per your initial suggestion, we have run our experiments on a different, correlated distribution and observed similar trends. Even though the distribution is globally not skewed, in any local neighborhood, most of the points belong to one color, according to our sampling procedure. Indeed, we partition the points into regions using random hyperplanes, and for each point within a region, choose a dominant color with 0.8 probability, and a uniformly random color out of 1000 colors with remaining probability. We see that our methods give good gains over the baseline approach.
>
> Results: We have the full plot for SIFT dataset, Max per color=1, which we cannot post as a comment here. However, e.g. at latency 4 our algorithm is at over 95% of Recall@20, whereas the baseline algorithm is around 60%. At latency 6, we are at 98% whereas the baseline is at 85%.

---

### Official Review · Reviewer_jfFS · 2024-11-05

**Soundness:** 3
**Presentation:** 2
**Contribution:** 2
**Rating:** 6
**Confidence:** 4

**Summary:**

This paper studies the topic of diverse approximate nearest neighbor (ANN) search and focuses on solving this problem via graph-based approaches as opposed to locality-sensitive hashing (LSH). LSH has been better studied in this context of finding diverse points in previous work, which lead to positive results, showing that diversity can be guaranteed at the same time when choosing points from hashed data structures. This paper takes a pioneering step towards finding the first algorithms for graph-based ANN search, which has become popular tools in the last decade or so but has not been considered from this perspective.

The paper studies this problem from both "primal" and "dual" formulations: one is to find a given $k$ diverse points optimizing closeness satisfying diversity constraint; the other to find most diverse points satisfying closeness constraints (within a ball of some radius $R$). The paper focuses on the primal but also showed similar results for the dual. The problem formulation allows approximation for full diversity: any point is allowed to have up to $k'$ points judged to be close to it. The main algorithm is relatively straight-forward. It starts with a preprocessing algorithm, which basically picks $k/k'$ points, that are far away from each other, each with its own set of close neighbors. This relationship is represented with a (sparse) graph. The algorithm then takes an initialization step that finds $k$ points that satisfies a much weaker diversity constraint, and finally compares the initialized set to the sparse graph and refine them to satisfy the $(k', C/\alpha)$-diverse constraint. In experiment section, the paper compares the new algorithm to the state-of-the-art graph-based ANN implementation and showed promising improvements.

**Strengths:**

The problem studied in this paper is both interesting and important in practice. The authors have motivated this very well. Also, adding diversity guarantees to graph-based approaches is a crucial step in bridging the gap between the two mainstream methods and might make graph-based approaches more applicable, since it can be used for more general metrics compared to LSH.

It is nice observation that the greedy algorithm in Gonzalez (1985), combined with an initialization, can lead to desirable results in diversity guarantees. The algorithm should also be easy to implement and likely could be used by practitioners.

At least in the described experiment setting, the algorithm beats its major baseline algorithm, which makes it more promising.

**Weaknesses:**

The writing appears messy in many places. It can benefit from some proofreading. For instance, some notions such as OPT, OPT_k appeared without being defined in advance. The lemmas and propositions in Section 3 are ordered and referred to in weird order, and the flow of logic in that section is not very smooth.

I haven't checked all the details, but the techniques used do not seem to be deep. In some places it feels that more sophisticated approaches might lead to better results. For example, I'm slightly disappointed that allowing a factor of $k'$ fuzziness in the diversity constraint only serves to reduce the running time linearly in $k'$. It makes sense to me judging from the way the algorithm works. However, in traditional ANN approaches, allowing approximation in distance can help us improve tremendously compared to exact nearest neighbor.  I'm not sure if similar improvements could be possible here.

The experiment design is more or less limited. The authors only uses the binary distance metric. I think it would be better to test it with other common graph-based metric that is not Euclidean. I'm also curious about how fast the running time scales with high dimensions in practice.

**Questions:**

Do the authors have any thoughts on how it would compare to LSH-based approaches, say, in Euclidean space using $\ell_2$ distance? It is unclear to me since this algorithm has exponential dependency on $d$.

I'm not sure if the $k'$-colorful design for distance metric is commonly used as a benchmark for this line of work, in some places it seems strange to me that if the distance is binary, the approximation in distance just disappears? In this case when $k'=1$ does it just become the nearest neighbor problem? If so, how does the running time of the algorithm given in this paper compare to the best you can do in that scenario?

---

> ### Author Response · Authors · 2024-11-20
> **Author Response to Reviewer jfFS -- 1**
>
> We thank the reviewer for helpful feedback. Below, we address specific comments and questions.
>
> - *“The writing appears messy in many places.”*
>
> Apologies if the writing was unclear. We will fix the $OPT_k$ confusion (we defined $S(q)_k$ but did not extend this definition to $OPT_k$). Please let us know any other comments.
>
> - *“I'm slightly disappointed that allowing a factor of k′ fuzziness in the diversity constraint only serves to reduce the running time linearly in $k'$.”*
>
> For the case when $k=k’=1$, the diverse (approximate) nearest neighbor problem specializes to the regular (approximate) nearest neighbor. In this case the runtime bounds of our algorithm  are qualitatively the same as the bounds for the regular approximate nearest neighbor in Indyk-Xu’23, but with a constant $8$ instead of $4$. Thus, we think it is unlikely that one could get a much larger improvement (say, polynomial in $n$) for other values of $k’$, unless the bounds for the regular version of the problem are improved as well. We also note that relaxing $k’$ to be greater than $1$ plays a very different role from the distance-based relaxation in approximate nearest neighbor.
>
> - *“In this case when $k’=1$ does it just become the nearest neighbor problem?”*
>
> When the diversity metric is binary (as in the experimental section), the case of $k’=1$ imposes the constraint that each of the $k$ points returned by the data structure must have a different color. To give an example, consider the main application motivating this paper, where the set of points is a collection of product ads shown to the user, each coming from a specific seller (e.g. Amazon, Ebay, etc). Furthermore, the user can be shown up to $k$ ads. Then setting $k'=1$ means that we require all the $k$ ads that we show to the user to come from different sellers, i.e., no seller should appear more than once. On the other hand, setting $k'=k$ means that we do not have any diversity constraints on the sellers.
>
> In reality, setting $k'=1$ is stricter than necessary from a user’s perspective. This motivated the relaxation.
>
> - *“The experiment design is more or less limited. The authors only uses the binary distance metric.”*
>
> The case of binary diversity metric was in fact the main practical motivation for this paper - we wanted to design a fast algorithm that reports $k$ products such that no seller appears too often in the list. In this setting the diversity metric is binary, but the distance metric is Euclidean.
>
> - *“I'm also curious about how fast the running time scales with high dimensions in practice.”*
>
> The graph construction algorithm we use for the baseline index, as well as our diverse-build, scales linearly in the ambient dimension d of the dataset. This is because the algorithms use the distance metric as a black-box, and evaluating the Euclidean distance between two points takes time proportional to $d$.
>
> On the other hand, one could imagine more complex experiments varying the *intrinsic* dimension of data. However, executing such an experiment with real data is difficult, as it would require a sequence of related data sets with varying intrinsic dimensionality.
>
> - *“It is nice observation that the greedy algorithm in Gonzalez (1985), combined with an initialization, can lead to desirable results in diversity guarantees.”*
>
> Thank you. We want to clarify though that we use the greedy algorithm only as one component in the pruning process. At a high level, to find the neighbors of a node $p$, we partition the space into balls of various radii, and use greedy within each ball and only connect $p$ to those $k/k'$ points returned by greedy. We then need to verify that such a sparsification is sufficient for our search algorithm to converge to an approximately optimal solution. (shown in lemma 3.3, lemma 3.5, and proof of theorem 1.1).
>
> - *“I haven't checked all the details, but the techniques used do not seem to be deep.”*
>
> We agree that the final algorithms are relatively simple (which we view as a plus, as simple algorithms are easier to implement). However, we believe that the simplicity of the final algorithms might have obfuscated several challenges that needed to be overcome. Even the algorithm for the simple case of $k'=1$ and binary metric $\rho$ metric turned out to be not easy to design and analyze. In particular, coming up with the “right” notion of an iterative step in the search algorithm (lines 10…14) and quantifying the progress that such a step makes (Lemmas 3.3 and 3.5) took a fair amount of effort.  Additional effort was needed to generalize the algorithm to arbitrary diversity metrics $\rho$ and $k’>1$.

---

> > ### Author Response · Authors · 2024-11-20
> > **Author Response to Reviewer jfFS -- 2**
> >
> > - *“Do the authors have any thoughts on how it would compare to LSH-based approaches, say, in Euclidean space using $\ell_2$ distance?”*
> >
> > It has been widely observed that, even in the setting without diversity requirements, state-of-art graph algorithms are more efficient than LSH (see e.g., Aumuller et al.). This holds despite the fact that the running time bounds of those algorithms from Indyk-Xu’23 also have exponential dependence on the intrinsic dimension $d$. This could be either because those bounds are not tight, or because the intrinsic dimensionality of real data sets is very low (the latter point has been also explored in Aumuller et al.).
> >
> > Regarding the diverse setting, we contacted the authors of the diversity-LSH paper Abbar et al., but unfortunately, their code is no longer available. However, given the point above, we expect the gap between our diverse graph-based algorithm and diverse LSH to be similar to the gap in the non-diverse setting.
> >
> > Finally, we want to add that the prior work only solves the dual version of the problem whereas here we are the first to also solve the primal version of the problem (which captures our application to seller diversity).

---

> > > ### Author Response · Authors · 2024-12-01
> > >
> > > Dear Reviewer jfFS, we wanted to check if we have addressed all your concerns and comments? Could you kindly let us know so that we have a chance to respond before the deadline? Thank you very much.

---

> > > > ### Comment · Reviewer_jfFS · 2024-12-02
> > > > **Thank you**
> > > >
> > > > Lots of thanks to the authors for the efforts they have put into addressing my concerns. I have read them and they added to my understanding of the paper. My general impression of the work remains roughly the same, and I would like to keep my score.

---

### Meta-Review · Area_Chair_gJmR · 2024-12-15

**Metareview:**

The paper proposes graph based near-neighbor search algorithms that also satisfies certain diversity constraints. As argued by the reviewers, the proposed theoretical analysis is rather incremental here, and the experiments are not convincing enough. As such I do not recommend the paper for acceptance at its current stage.

**Additional Comments On Reviewer Discussion:**

Rebuttal led to some increase of scores, but not enough to change the outcome.

---

### Decision · Program_Chairs · 2025-01-22

Reject